# Audible pain squeaks can mediate emotional contagion across pre-exposed rats with a potential effect of auto-conditioning

Julian Packheiser[1,7], Efe Soyman [1,2,7], Enrica Paradiso[1,7], Frédéric Michon[1], Eline Ramaaker[1], Neslihan Sahin[1], Sharmistha Muralidharan[1], Markus Wöhr [3,4,5,6], Valeria Gazzola [1,8] & Christian Keysers [1,8 ✉]

Footshock self-experience enhances rodents' reactions to the distress of others. Here, we tested one potential mechanism supporting this phenomenon, namely that animals auto-condition to their own pain squeaks during shock pre-exposure. In Experiment 1, shock pre-exposure increased freezing and 22 kHz distress vocalizations while animals listened to the audible pain-squeaks of others. In Experiment 2 and 3, to test the auto-conditioning theory, we weakened the noxious pre-exposure stimulus not to trigger pain squeaks, and compared pre-exposure protocols in which we paired it with squeak playback against unpaired control conditions. Although all animals later showed fear responses to squeak playbacks, these were weaker than following typical pre-exposure (Experiment 1) and not stronger following paired than unpaired pre-exposure. Experiment 1 thus demonstrates the relevance of audible pain squeaks in the transmission of distress but Experiment 2 and 3 highlight the difficulty to test auto-conditioning: stimuli weak enough to decouple pain experience from hearing self-emitted squeaks are too weak to trigger the experience-dependent increase in fear trans-mission that we aimed to study. Although our results do not contradict the auto-conditioning hypothesis, they fail to disentangle it from sensitization effects. Future studies could tem-porarily deafen animals during pre-exposure to further test this hypothesis.

[1] Social Brain Lab, Netherlands Institute for Neuroscience, Amsterdam, the Netherlands. [2] Social Cognitive and Affective Neuroscience Lab, Koc University, Istanbul, Turkey. [3] Research Unit Brain and Cognition, Laboratory of Biological Psychology, Social and Affective Neuroscience Research Group, Faculty of Psychology and Educational Sciences, KU Leuven, Leuven, Belgium. [4] Leuven Brain Institute, KU Leuven, Leuven, Belgium. [5] Behavioral Neuroscience, Experimental and Biological Psychology, Faculty of Psychology, Philipps-University Marburg, Marburg, Germany. [6] Center for Mind, Brain and Behavior, Philipps-University Marburg, Marburg, Germany. [7] These authors contributed equally: Julian Packheiser, Efe Soyman, Enrica Paradiso. [8] These authors jointly supervised this work: Valeria Gazzola, Christian Keysers. ✉email: c.keysers@nin.knaw.nl

Empathy refers to the ability to understand and share the feelings of other individuals[1–3]. The understanding comprises the cognitive component of empathy whereas the sharing comprises the affective component of empathy, which, in its simplest form, is also known as emotional contagion. Emotional contagion describes the tendency for emotional states to transmit from one individual to another leading to a convergence of emotional states[4–6]. Emotional contagion has been documented to be widespread in the animal kingdom and is considered a precursor to empathy[7].

Research on emotional contagion in non-human animals has largely focused on the emotional contagion of negative affective states such as pain or fear[6]. In both mice and rats, several studies have demonstrated that behavior indicative of fearful states, such as freezing, can be triggered by witnessing a demonstrator in distress[8–15]. If the demonstrator is exposed to numerous strong shocks, even observers who have not been pre-exposed to shocks show some level of freezing while witnessing the demonstrator receive shocks[11,12,15,16]. However, the response is strengthened if the observer has previously been pre-exposed to footshocks[9,14,16]. Interestingly, while in naïve observers the transmission of fear appears to depend strongly on vision[11,14], the pre-exposure appears to make animals more sensitive to auditory cues[13,14].

How the prior experience of footshocks can make animals respond more to the distress of others, in particular for auditory cues, remains incompletely understood. One mechanistic proposal that is particularly relevant for auditory cues is based on Hebbian learning and auto-conditioning[13,17–20]. When rats receive noxious footshocks, they emit two types of sounds. One type, called 'squeaks', can be heard by humans because they are broadband signals spanning across our audible and ultrasonic range. These squeaks are emitted at very short latency (<50 ms), their loudness reflects the intensity of the shock, and continue to be emitted in bouts for as long as the footshocks continue, i.e. for about 1 s when a 1 s footshock is delivered[21,22]. The other called 22 kHz ultrasonic vocalizations (USV), are beyond human hearing, are narrowband signals with a main frequency typically around 22 kHz, and are not emitted during the footshocks, but typically a few second after the shocks together with freezing[23,24]. Unlike the squeaks that are only emitted when rats are in very threatening or painful situations, USVs are emitted under many situations in which rats are in negative affective states, such as predator exposure, aggressive encounters, or long-lasting social isolation—in the laboratory they are typically seen in response to air puffs, acoustic startle stimuli, drug withdrawal, or footshocks[23]. From an information theoretical point of view, squeaks occur consistently during footshocks, yet rarely in other circumstances, making them a potentially highly informative cue about the pain state of others. From a neural perspective, that rats emit squeaks selectively while they experience footshocks also creates a Hebbian learning opportunity in which nociceptive neurons triggered by the footshocks are repeatedly co-activated with auditory neurons triggered by hearing themselves squeak, and the synapses connecting them could therefore be strengthened to the point where later hearing a demonstrator squeak would reactivate shock engrams and the associated nocifensive behaviors more strongly—explaining why pre-exposure could increase the reaction to other individuals being exposed to similar footshocks[18,19].

From a classical learning perspective, the shock is an unconditioned stimulus (US), hearing themselves squeak a conditioned stimulus (CS), and the US–CS contingency is a simultaneous conditioning protocol. Classic studies have demonstrated that roughly simultaneous tone-shock presentations may not be ideal, but can suffice to trigger nocifensive responses upon later presentations of the tone[25]—although effects are less strong than if the CS precedes and thus predicts the US—and the word 'auto-conditioning' refers to the fact that if the CS is a squeak produced by the animal itself, rather than a tone presented artificially by an experimenter, the animal essentially conditions itself to learn to fear the sounds it has produced in response to a threat[13,17,20]. Together these considerations beg us to test the notion that (a) hearing the squeaks of other animals may suffice to trigger fear responses in a pre-exposed listener and (b) that the contingency between a painful experience and hearing squeaks is what enhances later responses to hearing the squeaks of others, as Hebbian learning and auto-conditioning would suggest. Such auto-conditioning must be contrasted against a simpler, non-associative explanation based on sensitization alone: footshocks can increase nocifensive reactions to many stimuli independently of a specific contingency between a US and CS[26].

To our knowledge, this has not been tested for squeaks. Evidence that rodents may auto-condition in principle, stems from two lines of research that have explored auto-conditioning to other audible reactions to receiving footshocks: freezing and USVs. Regarding freezing, when shock pre-exposed observer rats hear that another rat stops moving, the observer freezes[27]. However, only rats that had experienced the shock event in combination with their own freezing showed fear when later hearing the cessation of motion sounds—those that were prevented from freezing did not. This suggests that auto-conditioning via the association of their own freezing and fear, and not sensitization due to the shocks, were responsible for the potentiation of the freezing when hearing silence[16]. Regarding USVs that often accompany freezing behavior[24], naïve rats do not respond specifically with fear when hearing these calls[28] but see Parsana et al.[29]. However, rats pre-exposed to shocks do later freeze in response to demonstrators emitting 22 kHz or to playbacks of 22 kHz vocalizations[13,30]. Whether this was due to auto-conditioning however remains somewhat unclear, as Kim et al.[13] find that preventing rats from hearing themselves produces USVs during pre-exposure by deactivating the auditory thalamus reduces later responses to USVs, while Calub et al.[30] find that preventing rats from producing USVs during pre-exposure, using a devocalization surgery, does not reduce later responses to USV playbacks. It is worth noting, that the fact that USVs are not emitted during shocks, but a couple of seconds after the shocks, again creates an unfavorable situation for auto-conditioning, as protocols in which a tone follows a shock (so-called backward conditioning paradigms) sometimes even lead to safety learning as the CS gains inhibitory value predictive of a US-free intertrial interval (e.g., Heth and Rescorla[25]).

Auto-conditioning is thus generally plagued by unfavorable timings of the self-emitted CS relative to the US. Can auto-conditioning thus happen at all? Several studies have found that a CS can gain excitatory value even if it starts after the US, be it in a simultaneous or even backward conditioning configuration (e.g., Barnet et al.[31]; Cole and Miller[32]; Prével et al.[33]), although the boundary conditions for simultaneous or backward conditioning to occur are likely more constrained compared to forward conditioning. For a long time, this seemed at odds with the basic tenets of learning theory, that stimuli need to be predictive to trigger learning (for an overview, see Chang et al.[34]). A recent study has started to shed light on the neural mechanisms of learning in situations in which the CS is not temporally predictive of the US, by showing that the dopaminergic system, important for forward conditioning, is also involved in such backward conditioning[35].

Given that responses to USVs, and to the silence accompanying freezing have already been explored to some depth, while responses to squeaks have not, and given the rather unique contingency between the timing of footshocks and squeaks, here

we will focus on the effect of pre-exposure on the response to squeak playbacks, the specificity for squeaks, and whether the pairing of squeaks and shocks during pre-exposure is important. In Experiment 1 we investigate the role of shock pre-exposure on fear responses during an auditory squeak playback test and compare the responses to those of a phase-scrambled version of squeaks to assess specificity. Whether pre-exposure acts via auto-conditioning or sensitization, we expect rats to respond more strongly to squeak playbacks after they have been pre-exposed to shocks compared to shock-naive animals. However, auto-conditioning but not sensitization would predict that responses should be particularly increased for the intact squeaks (that they heard themselves emit during pre-exposure) compared to phase scrambled squeaks (that they never heard themselves produce). In Experiments 2 and 3 we attempt to experimentally manipulate the contingency between the noxious experience during pre-exposure and the sound of squeaking. In Experiment 2 we did so by using a $CO_2$ laser instead of a footshock, which triggers paw-retractions indicative of pain, but no squeaking. By aligning the playback of a squeak to the administration of the laser in one group but not the other, we then aimed to study the importance of squeak-pain conditioning against mere sensitization. In Experiment 3 we did so by reducing footshock intensity to a level at which rats seldom emit squeaks during pre-exposure, and again aimed to study the importance of squeak-pain conditioning against mere sensitization by comparing individuals in which we synchronize a squeak playback to the mild shock administration against a group where we do not.

## Results

**Experiment 1: Pre-exposure session.** All animals in the Shock → Squeak and Shock → Control groups (19/19) emitted audible squeak vocalizations in response to each of the four footshocks, whereas none of the animals in the NoShock → Squeak group (0/5) emitted any squeaks during their sham pre-exposure session. Freezing differed between the experimental groups (S–W p-value = 0.211; $F_{(2,21)} = 100.01$, p < 0.001, $\eta^2 = 0.91$, $BF_{incl} > 100$) and was higher in both the Shock → Squeak and Shock → Control group compared to the NoShock → Squeak group (both ps < 0.001, $BF_{10}s > 100$, see Fig. 1a). The frequentist statistic indicates that there was no difference between the two groups that received shocks and the Bayesian statistic, while not yet indicating evidence in support of the lack of a difference, indicates a trend toward a similar conclusion (p = 1.000, $BF_{10} = 0.52$). This was also the case for 22 kHz recordings in the ten animals for which USV recordings were available ($t_{(8)} = 1.08$, p = 0.310, $BF_{10} = 0.70$, see Fig. 1b).

Descriptive statistics for freezing responses and 22 kHz vocalizations during pre-exposure are presented in Supplementary Tables 1 and 2.

To provide further insights into the timing of the behavioral responses of the animals to the pre-exposure shocks, we analyzed, for each 1 s interval relative to the onset of the shocks, what proportion of trials included squeaks, 22 kHz vocalizations or freezing (Fig. 2). We found that all animals emitted squeaks in all trials during the second of shock delivery, but never outside of that epoch. This tight contingency between shocks and squeaks would be ideal for threat conditioning. In contrast, 22 kHz vocalizations occurred outside of the shock delivery period and increased significantly 2 s after shock delivery. Freezing was highest before shock delivery, with shock delivery reducing freezing as it triggers more active forms of defensive behavior. This data stems from averaging four pre-exposure shocks for each animal so that the 5 s preceding shocks combine a true baseline period preceding the first shock with the inter-shock interval preceding the other three shocks. The freezing rate of around 75% prior to shock delivery reflects this averaging of ¼ trials with virtually no freezing during the baseline period and ¾ of trials with high freezing after the preceding shock.

**Experiment 1: Playback session.** First, we aimed to verify that fear responses were comparable between the experimental groups during the baseline period of the playback session (Fig. 3). The experimental groups did not show differences in freezing responses (S–W p-value = 0.128; $F_{(2,20)} = 2.46$, p = 0.110, $\eta^2 = 0.19$) or in 22 kHz call emissions (S–W p-value < 0.001; $\chi^2_{(2)} = 1.31$, p = 0.519, $\eta^2 = 0.03$). These results were complemented by corresponding Bayesian ANOVAs which indicated either the absence of evidence of an effect for freezing responses ($BF_{incl} = 0.69$) or moderate evidence of absence for the 22 kHz vocalizations indicative of distress ($BF_{incl} = 0.30$).

Next, we aimed to determine differences in fear behavior during the playback period across groups (Fig. 4). We therefore compared the freezing rates and 22 kHz vocalizations between the three experimental groups using a one-way ANOVA expecting higher fear responses in the Shock → Squeak group in line with the auto-conditioning hypothesis. We found significant main effects on both freezing rates (S–W p-value = 0.608; $F_{(2,20)} = 4.32$, p = 0.028, $\eta^2 = 0.30$, $BF_{incl} = 2.11$, Fig. 3a) as well as 22 kHz call emissions (S–W p-value = 0.221; $F_{(2,20)} = 8.39$, p = 0.002, $\eta^2 = 0.45$, $BF_{incl} = 9.67$, Fig. 3b). In contrast to our hypothesis, planned comparisons for freezing rates showed absence of evidence for a difference between the Shock → Squeak and the Shock → Control group ($t = 0.50$, p = 0.601, $BF_{10} = 0.64$

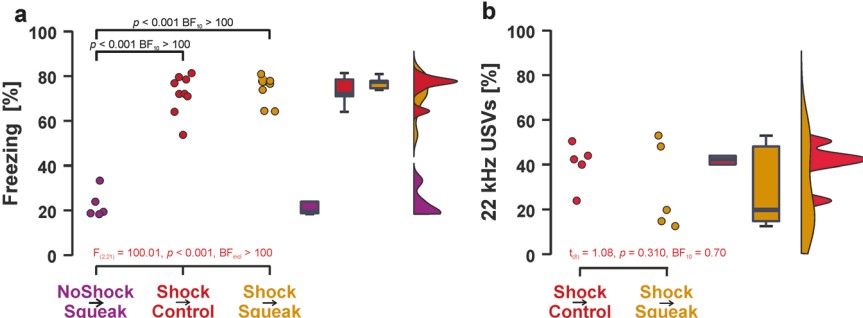

**Fig. 1 Freezing responses and 22 kHz vocalizations during pre-exposure. a** Animals that received shocks during pre-exposure (red and yellow) showed higher freezing responses during pre-exposure than animals not receiving shocks (lilac). Freezing was comparable in magnitude for the two shock groups. Freezing was analyzed for all 24 animals tested in Experiment 1. **b** 22 kHz vocalizations for animals that received shocks during pre-exposure. Note that USVs were recorded for five animals from the Shock → Control and five animals from the Shock → Squeak group only, and no USVs were available for the NoShock group. Boxplots reflect the median, the first and third quartile, and the interquartile range.

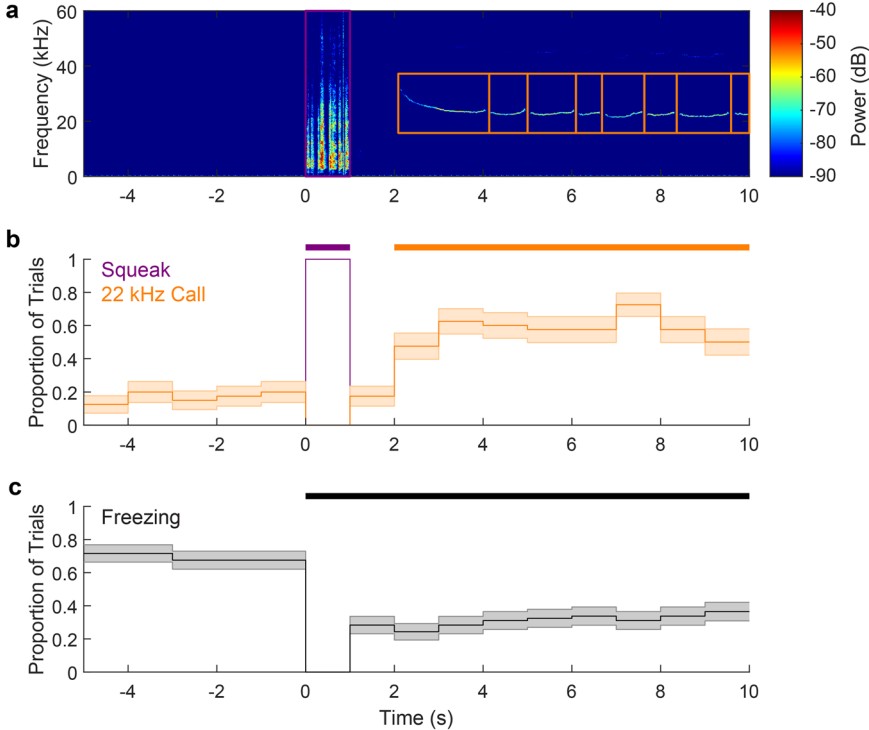

**Fig. 2 Behavioral responses relative to the timing of footshocks during pre-exposure for Experiment 1. a** Exemplar spectrogram in response to the second shock an animal received during the pre-exposure session. The lilac box highlights the squeak occurring during the 1 s of shock delivery, while the orange boxes highlight eight 22 kHz vocalizations. Squeaks only occurred during shocks, whereas 22 kHz calls were absent during shocks. **b** Proportions of trials during which 22 kHz calls (orange) or squeaks (lilac) were produced as a function of 1 s bins from 5 s before until 10 s after the shock onset. Note that shocks start at $t = 0$ and last for 1 s. For each trial, if more than 0.5 s of a particular bin contained a 22 kHz call or a squeak, that bin was scored as 1, if not, scored as 0. Note that the squeaks and 22 kHz calls were only recorded from the subset of 10 animals that had a lower amplitude playback during the playback session, as we had not initially planned to analyze (and hence record) sound emissions during pre-exposure. There were four shocks for each of these 10 animals; thus, the proportions were calculated across $10 \times 4 = 40$ total shock trials. Given that 4 shocks are given per animal, the period from $-5$ to 0 s is not a true baseline, but an interval that in 3/4 of cases occurs after another shock (the inter-shock interval is either 240 or 360 s). Error bars indicate standard error of proportions (SEP) as calculated separately for each bin as $SEP = (p(1-p)/n)^{1/2}$, where $p$ is the observed proportion and $n$ is the total number of shock trials. The thick orange bar above the figure indicates significant increases in 22 kHz call emission compared to the baseline period as analyzed by comparing each of the observed proportions after shock onset to the average of the proportions observed during the 5 s baseline period via separate binomial tests (all $p$s < 0.005). Squeaks were observed in all shock trials for all animals during the 1 s of footshock delivery, but never outside that bin. **c** Same as in **b**, but for freezing responses. Note that freezing analysis was conducted across all 24 animals. The thick black bar above the figure indicates significant changes in freezing due to shock exposure.

suggesting an unspecific fear response following shock pre-exposure. There was approaching moderate evidence that freezing was increased in the Shock → Squeak compared to the NoShock → Squeak group ($t = 2.56$, $p = 0.025$, $BF_{10} = 2.85$). For 22 kHz calls, we however found approaching moderate evidence for the Shock → Squeak group emitting more 22 kHz calls compared to the Shock → Control group ($t = 2.41$, $p = 0.014$, $BF_{10} = 2.55$) and moderate evidence to emit more 22 kHz calls compared to the NoShock → Squeak group ($t = 3.16$, $p = 0.004$, $BF_{10} = 6.26$) in line with the auto-conditioning hypothesis. Playback amplitude neither reached significance for freezing ($F_{(1,20)} = 0.10$, $p = 0.752$) nor 22 kHz calls ($F_{(1,20)} = 0.47$, $p = 0.500$) suggesting that it did not play a role in modifying the fear responses, a result contrasting a previous study on locomotor inhibition[36].

Linear regression analyses revealed that freezing during pre-exposure was predictive of freezing during the playback ($F_{(1,22)} = 6.26$, $p = 0.020$, $BF_{10} = 3.08$). This was not the case for 22 kHz vocalizations, however ($F_{(1,8)} = 1.32$, $p = 0.283$, $BF_{10} = 0.73$). There was no significant correlation between the emission of 22 kHz vocalizations and freezing responses during squeak playback ($r_{(23)} = 0.33$, $p = 0.117$, $BF_{10} = 0.89$, Spearman correlation).

Results for each individual experimental group are depicted in Fig. 3a for freezing and in Fig. 3b for 22 kHz calls. Descriptive statistics for freezing responses and 22 kHz calls are presented in Supplementary Tables 3 and 4, respectively. A time-resolved figure representing freezing responses and 22 kHz calls for each stimulus presentation can be found in Supplementary Fig. 1. A comparison between baseline and playback during the playback session can be found in Supplementary Figs. 2 and 3 for freezing and 22 kHz vocalizations, respectively.

To account for inter-individual variability in baseline fear responses, we repeated the previous analyses and used the relative increase from baseline to playback as the dependent variable. The overall result pattern was identical to a non-baseline corrected analysis as we again found significant main effects and moderate evidence that freezing rates (S–W $p$-value = 0.657, $F_{(2,20)} = 6.14$, $p = 0.008$, $\eta^2 = 0.34$, $BF_{incl} = 4.71$) and strong evidence that 22 kHz call emissions (S–W $p$-value = 0.015; $\chi^2_{(2)} = 9.38$, $p = 0.007$, $\eta^2 = 0.37$, $BF_{incl} = 21.02$) differed between the groups. As for the data from the playback period only, there was the absence of evidence that the Shock → Squeak group differed in terms of freezing compared to the Shock → Control group ($t = 1.38$, $p = 0.149$, $BF_{10} = 0.61$). Similarly, there was absence of evidence for a difference in freezing between the

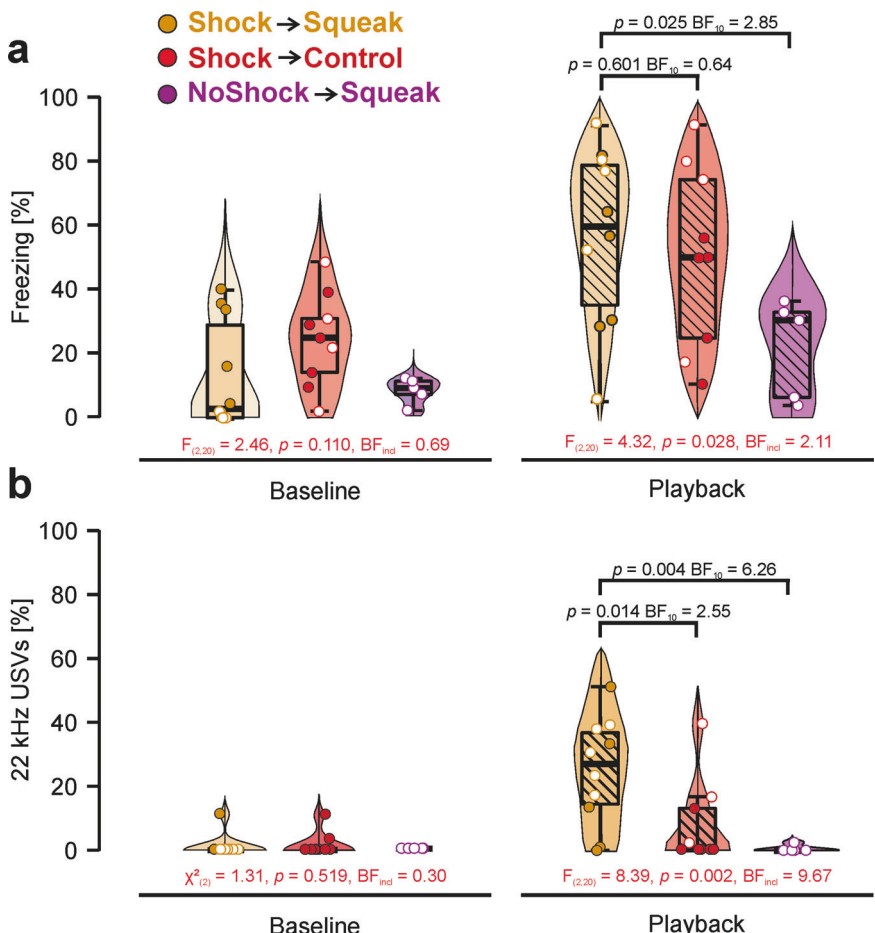

**Fig. 3 Behavioral results for the playback session on day 5. a** Proportion of freezing responses in percent during baseline and auditory playback in the playback session. Animals with higher amplitude playback are marked by an open circle. Only between-group differences are presented here. Within-group differences from baseline to playback are depicted in Supplementary Fig. 2. **b** Proportion of 22 kHz calls in percent during baseline and auditory playback in the playback session. Animals with higher amplitude playback are marked by an open circle. Only between-group differences are presented here. Within-group differences from baseline to playback are depicted in Supplementary Fig. 3. *Represents $p < 0.05$ and ** represents $p < 0.01$. Boxplots reflect the median, the first and third quartile, and the interquartile range.

Shock->Squeak compared to the NoShock → Squeak group ($t = 1.93$, $p = 0.045$, $BF_{10} = 1.38$). Amplitude as a covariate did not reach significance for freezing rates ($F_{(1,20)} = 3.81$, $p = 0.065$). For 22 kHz calls and in line with our prediction that fear responses should be specific to squeak playback, we found a significant effect and anecdotal evidence for an increase for the Squeak → Shock group compared to the NoShock → Squeak group ($U = 4$, $p = 0.009$, $BF_{10} = 1.94$). The comparison to the Shock → Control group was significant but revealed absence of evidence ($U = 17.5$, $p = 0.025$, $BF_{10} = 1.49$).

The final analysis we conducted aimed to identify whether squeak playback elicited similar freezing responses compared to a classical emotional contagion design during which the observer is paired with another conspecific. Here, we made use of pre-existing data of nine rats from a previously published study[16] and compared them to the data from the Shock → Squeak group. The direct comparison between the result patterns of these two studies is possible since the experimental protocol (pre-exposure: session duration, number of shocks, shock amplitude, and interstimulus interval || test session: session duration, number of shocks/squeak playbacks) of the present study was identical to the study by Han et al.[16]. 22 kHz calls were not compared between the studies as they were not recorded in Han et al.[16]. While the freezing rates were reduced on a descriptive level during squeak playback ($56.69 \pm 27.95\%$) compared to a classical observation

($69.68 \pm 14.57\%$), an independent sample $t$-test showed absence of evidence for or against a group difference (S–W $p$-values > 0.310; $t_{(16)} = 1.25$, $p = 0.229$, $d = 0.57$, $BF_{10} = 0.69$). For a baseline-corrected measure of freezing, a similar picture emerged (S–W $p$-values > 0.300; $t_{(16)} = 1.39$, $p = 0.182$, $d = 0.64$, $BF_{10} = 0.78$, Supplementary Fig. 4).

In the first experiment, we investigated the auto-conditioning hypothesis in the context of pain squeaks. According to the auto-conditioning hypothesis, the self-emitted squeaks become a CS that associates with the shocks as the US during pre-exposure. Hearing these squeaks in conspecifics then later triggers the conditioned response. We tested whether rats become auto-conditioned to squeaking after having the opportunity to associate squeaks with aversive experiences during pre-exposure to painful shocks. As expected, freezing and 22 kHz call rates increased for the shock pre-exposed animals from the baseline to the playback period, and were generally higher for the Shock → Squeak compared to the NoShock → Squeak group during the playback period. Although only as a trend, freezing responses tended to increase to the presentation of squeaks compared to baseline also in the NoShock → Squeak condition (Supplementary Fig. 2). Playing back the scrambled squeaks elicited a similar level of freezing as regular squeak playback but lower levels of 22 kHz calls. Such responses indicate that not only auto-conditioning (that should be more specific to the self-emitted

intact squeaks) but also sensitization to auditory stimuli more generally (including the phase-scrambled squeaks) might have been at play after being exposed to aversive stimuli. Sensitization to fear occurs after a harmful and possibly traumatic event that causes subsequently elevated fear and stress responses under conditions that would normally not trigger such responses[37]. Altogether these results suggest auto-conditioning may not be necessary to respond to the distress of others but could act as an enhancer of an innate disposition to react to squeaks.

To disambiguate if the underlying mechanism facilitating emotional contagion was due to sensitization or auto-conditioning to squeaks (or possibly even both), the squeak and the aversive event need to be disentangled. To this end, in Experiment 2 we substituted the footshocks during pre-exposure, which unfortunately trigger both pain and the emission of pain squeaks, with a painful experience that has been demonstrated not to elicit squeaking: shining a $CO_2$ laser on the animal[22]. By pairing this painful laser experience with or without a squeak playback, auto-conditioning and sensitization should be separable: If the fear responses to later squeak playback are due to auto-conditioning, freezing rates and 22 kHz call emissions should only be increased in an experimental group that received paired pain pre-exposure with squeak playback (auto-conditioning) compared to a group with pain pre-exposure but without squeak playback pairing (sensitization). If fear responses are due to sensitization alone, both groups should show equal levels of freezing and 22 kHz calls. If both mechanisms play a role as hinted at in Experiment 1, both the auto-conditioning and the sensitization group should demonstrate stronger fear responses compared to controls, and the auto-conditioning group should display stronger fear responses compared to the sensitization group.

**Experiment 2: Pre-exposure (day 2).** In the first step, we compared responses to our customized pain reaction scale between all experimental groups that either received a high (Laser and Laser + Squeak) or low-intensity laser stimulation (Squeak and Naive; Fig. 4a). Because the most frequently used criterion to determine whether a stimulus is above the pain threshold is to determine whether a given trial did or did not lead to paw withdrawal, and to consider stimuli that trigger such withdrawal in at least 50% of trials to be above pain threshold[38], we also analyzed our data in terms of the number of trials (out of the possible 4, Fig. 4b), in which each animal withdrew their paw. A one-way ANOVA revealed a highly significant difference across groups ($F_{(3,68)} = 211.02$, $p < 0.001$, $BF_{10} > 100$), with post hoc tests revealing that this was due to the two conditions with Laser triggering a similar number of withdrawals ($t = 1.91$, $p = 0.23$, $BF_{10} = 1.09$) that was significantly higher than that for the two conditions without laser (all $ts > 16.80$, all $ps < 0.001$, all $BF_{10}s > 100$), which in turn were similar ($t = 0.30$, $p = 0.990$, $BF_{10} = 0.34$). Importantly, the median number of pain-like behavioral responses (as defined in the literature as paw-withdrawal) was zero for the two conditions without laser, and 4 for the conditions with laser. If one uses a 50% response threshold to determine whether a stimulus intensity was above or below the pain threshold, this provides strong evidence that the application of the laser did trigger pain in the Laser or Laser + Squeak groups but not in the Naïve or Squeak only groups. Whether the pain level was similar to that triggered by footshocks is doubtful, as pain squeaks, considered evidence for relatively intense pain[39], were not observed in any of the animals during pre-exposure in Experiment 2. Descriptive statistics for pain responses can be found in Supplementary Table 5.

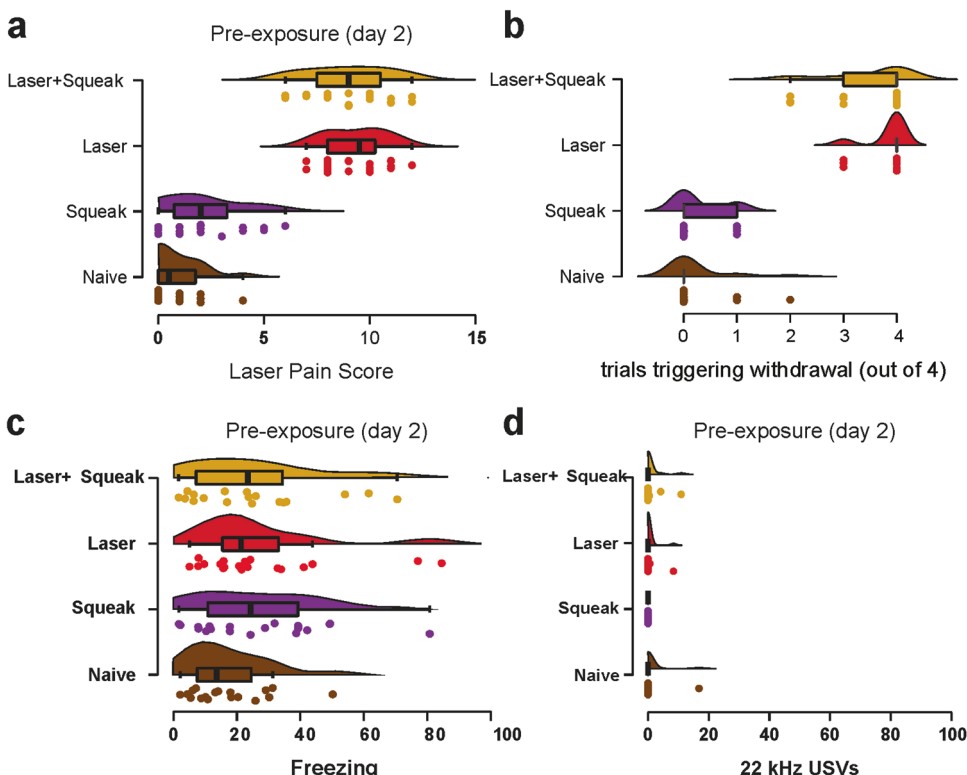

**Fig. 4 Pain and fear responses during pre-exposure on day 2. a** Cumulative pain responses to the laser for the four experimental groups (left), as well as the number of trials (amongst the four for each animal) that triggered paw withdrawal or a full-body escape. Descriptive values for pain scores can be found in Supplementary Table 5. **b** As in **a** but illustrating the number of trials inducing withdrawal. **c** Proportion of freezing and **d** proportion of 22 kHz calls for the different experimental groups. Boxplots reflect the median, the first and third quartiles, and the interquartile range.

We also quantified freezing rates as well as 22 kHz vocalizations during laser pre-exposure to identify the laser-induced discomfort in the animals. Unlike the clear pain reactions observable during high-intensity laser exposure, we found moderate evidence of absence that the groups differed in freezing (S–W $p$-value $< 0.001$; $\chi^2_{(3)} = 3.06$, $p = 0.382$, $\eta^2 < 0.01$, $BF_{incl} = 0.23$, Fig. 4c) as well as strong evidence of absence that the experimental groups emitted different levels of 22 kHz calls (S-W $p$-value $< 0.001$; $\chi^2_{(3)} = 3.41$, $p = 0.322$, $\eta^2 < 0.01$, $BF_{incl} = 0.07$, Fig. 4d). No animals emitted any squeaks over the course of pre-exposure. Descriptive statistics for freezing responses and 22 kHz call emissions during pre-exposure are presented in Supplementary Tables 6 and 7, respectively.

**Experiment 2: Squeak playback (day 5).** During the squeak playback session on day 5, we again first investigated baseline differences in fear responses to ensure that there was no difference in basal fear levels between the experimental groups. We found moderate evidence against differences between the experimental groups for freezing (S–W $p$-value $< 0.001$; $\chi^2_{(3)} = 1.06$, $p = 0.788$, $\eta^2 < 0.01$, $BF_{incl} = 0.16$) and 22 kHz calls (S–W $p$-value $< 0.001$; $\chi^2_{(3)} = 4.05$, $p = 0.256$, $\eta^2 = 0.01$, $BF_{incl} = 0.24$). Contrary to our expectations, there was also anecdotal to moderate evidence against group differences during the playback session (freezing: S–W $p$-value $< 0.001$; $\chi^2_{(3)} = 3.13$, $p = 0.373$, $\eta^2 = 0.01$, $BF_{incl} = 0.16$; 22 kHz calls: S–W $p$-value $< 0.001$; $\chi^2_{(3)} = 5.89$, $p = 0.117$, $\eta^2 = 0.04$, $BF_{incl} = 0.40$) as well as for the difference scores between playback and baseline (freezing: S–W $p$-value $< 0.001$; $\chi^2_{(3)} = 3.45$, $p = 0.328$, $\eta^2 = 0.01$, $BF_{incl} = 0.15$; 22 kHz calls: S–W $p$-value $< 0.001$; $\chi^2_{(3)} = 6.63$, $p = 0.08$, $\eta^2 = 0.05$, $BF_{incl} = 0.41$). Results for each individual experimental group are depicted in Fig. 5a for freezing and Fig. 5b for 22 kHz calls.

Using linear regression, we found that freezing during pre-exposure was not predictive of freezing during the playback ($F_{(1,73)} = 1.21$, $p = 0.275$, $BF_{10} = 0.40$). The same was true for 22 kHz vocalizations across animals ($F_{(1,75)} = 1.04$, $p = 0.311$, $BF_{10} = 0.37$). There was a significant correlation between the emission of 22 kHz vocalizations and freezing responses during squeak playback ($r_{(76)} = 0.46$, $p < 0.001$, $BF_{10} > 100$, Spearman correlation).

Descriptive statistics for freezing rates and 22 kHz calls during normal squeak playback are presented in Supplementary Tables 8 and 9, respectively. A time-resolved figure representing freezing responses and 22 kHz calls for each stimulus presentation can be found in Supplementary Fig. 5. A comparison between baseline and playback during the playback session can be found in Supplementary Figs. 6 and 7 for freezing and 22 kHz vocalizations, respectively.

**Experiment 2: Phase-scrambled squeak playback (day 7).** During the phase-scrambled squeak playback session on day 7, we replicated the analysis procedure for the normal squeak playback. As before, we found anecdotal to strong evidence against differences between the experimental groups for freezing (S–W $p$-value $< 0.001$; $\chi^2_{(3)} = 0.69$, $p = 0.788$, $\eta^2 < 0.01$, $BF_{incl} = 0.13$) and 22 kHz calls (S–W $p$-value $< 0.001$; $\chi^2_{(3)} = 2.55$, $p = 0.466$, $\eta^2 = 0.01$, $BF_{incl} = 0.70$) during the baseline. Similarly, there was moderate to strong evidence against group differences during the playback session (freezing: S–W $p$-value $< 0.001$; $\chi^2_{(3)} = 0.30$, $p = 0.961$, $\eta^2 < 0.01$, $BF_{incl} = 0.09$; 22 kHz calls: S–W $p$-value $< 0.001$; $\chi^2_{(3)} = 2.00$, $p = 0.573$, $\eta^2 = 0.01$, $BF_{incl} = 0.11$) and for the difference scores between playback and baseline (freezing: S–W $p$-value $< 0.001$; $\chi^2_{(3)} = 0.60$, $p = 0.896$, $\eta^2 < 0.01$, $BF_{incl} = 0.09$; 22 kHz calls: S–W $p$-value $< 0.001$; $\chi^2_{(3)} = 3.31$,

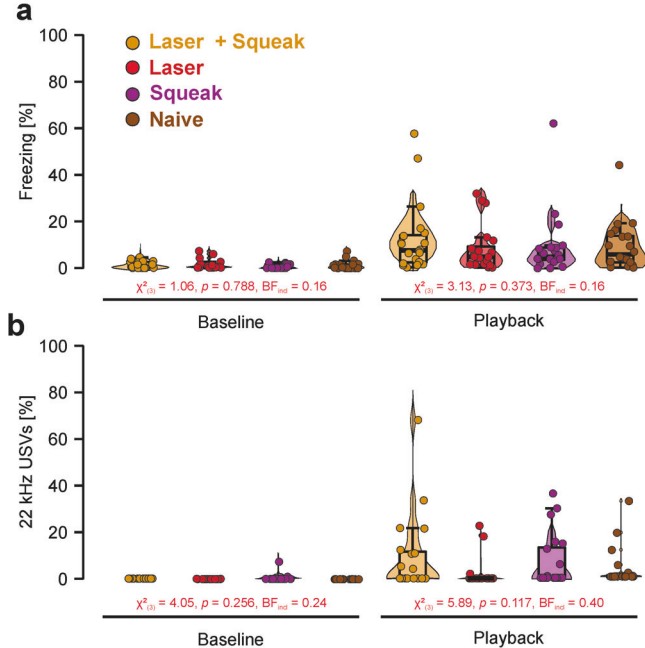

**Fig. 5 Behavioral results for the squeak playback (day 5) of Experiment 2. a** Proportion of freezing responses in percent during baseline and auditory playback in the playback session on day 5. Animals with higher amplitude playback are marked by an open circle. Only between-group differences are presented here. Within-group differences from baseline to playback are depicted in Supplementary Fig. 6. **b** Proportion of 22 kHz vocalizations in percent during baseline and auditory playback in the playback session on day 5. Animals with higher amplitude playback are marked by an open circle. Only between-group differences are presented here. Within-group differences from baseline to playback are depicted in Supplementary Fig. 7. Boxplots reflect the median, the first and third quartile, and the interquartile range.

$p = 0.346$, $\eta^2 < 0.01$, $BF_{incl} = 0.11$). Results for each individual experimental group are depicted in Supplementary Fig. 8. The corresponding within-subject baseline-playback comparisons are depicted in Supplementary Figs. 9 and 10, respectively. Descriptive statistics for freezing rates and 22 kHz calls during phrase-scrambled squeak playback are presented in Supplementary Tables 10 and 11, respectively.

**Cross-experimental comparison of fear responses during squeak playback.** In an exploratory analysis, we investigated whether freezing differed between Experiments 1 and 2 during playback in groups with aversive pre-exposure. To this end, we conducted a Wilcoxon rank sum test for freezing responses and 22 kHz calls during squeak playback between shock or laser-pre-exposed animals from both experiments. Fear responses were pooled across both groups that received laser pre-exposure as there was evidence against group differences in the previous analysis. We found moderate evidence for increased freezing (S–W $p$-value $< 0.001$; $W = 37.5$, $p < 0.001$, $BF_{10} = 5.05$) and increased 22 kHz calls (S–W $p$-value $< 0.001$; $W = 58.5$, $p < 0.001$, $BF_{10} = 3.16$) after shock pre-exposure compared to laser pre-exposure.

In the second experiment, we aimed to disentangle the effect of the potential association created by pairing self-experience with painful events and squeak emissions on subsequent fear responses during squeak playbacks. We hypothesized that the fear responses would be increased upon playback if pain pre-exposure in combination with squeak playback were given compared to a group exposed only to a pain stimulus. If fear responses were due

to sensitization, both groups would have shown equal levels of freezing and 22 kHz calls, but both would be higher than the pre-exposures not including pain. None of the hypotheses could be confirmed as neither freezing levels nor 22 kHz call emissions differed between the experimental groups that received a painful experience during pre-exposure and the controls without pain self-experience. These results were surprising given that $CO_2$ laser stimulation has been previously used to induce fear conditioning[40,41] and it has been shown to trigger emotional mirror neurons which are active during the observation of other rats receiving painful shocks[21]. We discuss these findings within the scope of the general discussion. In addition, although all groups showed increased freezing upon playback compared to baseline, this did not differ between the intact and scrambled squeaks and did not reach the levels observed following pre-exposure to footshocks.

A clear limitation of laser-induced pain is that a direct comparison between Experiments 1 and 2 is questionable despite the otherwise similar experimental protocols since the modality of the painful experience is very different between laser and shock exposure. Since we could also not find any differences between the experimental groups receiving laser pain and the control groups, we conducted a further experiment to disentangle the contribution of auto-conditioning and sensitization and increase the comparability to Experiment 1. Although exposure to shocks generally induces squeaks, the invariance of squeaking can be modulated through the intensity of the electrical stimulus. Thus, a new experiment was conceived using low-intensity shocks to induce an aversive experience without eliciting pain squeaks in the animals. To dissociate between auto-conditioning and sensitization effects, we used comparable procedures as in Experiment 2 by using squeak playback during shock pre-exposure.

**Experiment 3: Pre-exposure (day 2).** We again first compared fear responses between the Synchronous and Asynchronous groups during the mild shock pre-exposure. We found anecdotal to moderate evidence against a difference in freezing (S–W p-value < 0.001; $W = 38$, $p = 0.393$, $BF_{10} = 0.51$, Fig. 6a) as well as 22 kHz calls (S–W p-value = 0.008; $W = 41$, $p = 0.428$, $BF_{10} = 0.43$, Fig. 6b) during pre-exposure. In total, we presented 40 shocks to animals in the Asynchronous group ($n = 10$ animals × 4 shocks). The animals in the Asynchronous group squeaked in $11/40 = 27.5\%$ of shocks suggesting that the shock intensity could not have been any higher without compromising

the experimental manipulation. Squeaking in the Synchronous group could not be measured as they occurred simultaneous to the squeak playback. Descriptive statistics for freezing responses and 22 kHz call emissions during pre-exposure are presented in Supplementary Tables 12 and 13, respectively.

**Experiment 3: Squeak playback (day 5).** During the squeak playback session on day 5, we again first investigated baseline differences between the experimental groups in fear responses. We found an absence of evidence for freezing (S–W p-value < 0.001; $W = 74$, $p = 0.052$, $BF_{10} = 0.87$) and anecdotal to moderate evidence against a difference between groups for 22 kHz vocalizations (S–W p-value < 0.001; $W = 50$, $p = 1.000$, $BF_{10} = 0.40$). During the playback, we found anecdotal to moderate evidence against group differences for freezing (S–W p-value = 0.273; $t_{(18)} = 1.32$, $p = 0.203$, $BF_{10} = 0.51$) and absence of evidence for 22 kHz vocalizations (S–W p-value < 0.001; $W = 31.5$, $p = 0.093$, $BF_{10} = 0.72$). The same results were found for the difference scores between playback and baseline (freezing: S–W p-value = 0.592; $t_{(18)} = 0.83$, $p = 0.417$, $BF_{10} = 0.73$; 22 kHz calls: S–W p-value < 0.001; $W = 29$, $p = 0.069$, $BF_{10} = 0.88$).

To test whether animals for whom the pre-exposure might have been more distressing would later respond more intensely to playback, we examined whether there was an association between individual differences in behavior during pre-exposure and playback. Using linear regression, we found that freezing during pre-exposure was not predictive of freezing ($F_{(1,18)} = 0.26$, $p = 0.619$, $BF_{10} = 0.44$) during playback. The results were similar for 22 kHz vocalizations ($F_{(1,18)} = 0.58$, $p = 0.455$, $BF_{10} = 0.49$). Since we could measure squeaking during pre-exposure in the Asynchronous group and observed occasional squeaks, we correlated the number of emitted squeaks during pre-exposure with the fear responses during playback to identify if more squeaking during pre-exposure was associated with higher freezing responses or 22 kHz vocalizations upon playback. We did not find an association for either variable in the Asynchronous group (freezing: $r_{(9)} = -0.47$, $p = 0.173$, $BF_{10} = 0.94$; 22 kHz USVs: $r_{(9)} = 0.24$, $p = 0.501$, $BF_{10} = 0.56$, Spearman correlations). The correlation between the emission of 22 kHz vocalizations and freezing responses during squeak playback did not reach significance but showed anecdotal evidence in favor of the alternative hypothesis ($r_{(19)} = 0.37$, $p = 0.105$, $BF_{10} = 1.42$, Spearman correlation).

Results for each individual experimental group are depicted in Fig. 7a for freezing and Fig. 7b for 22 kHz calls. Descriptive

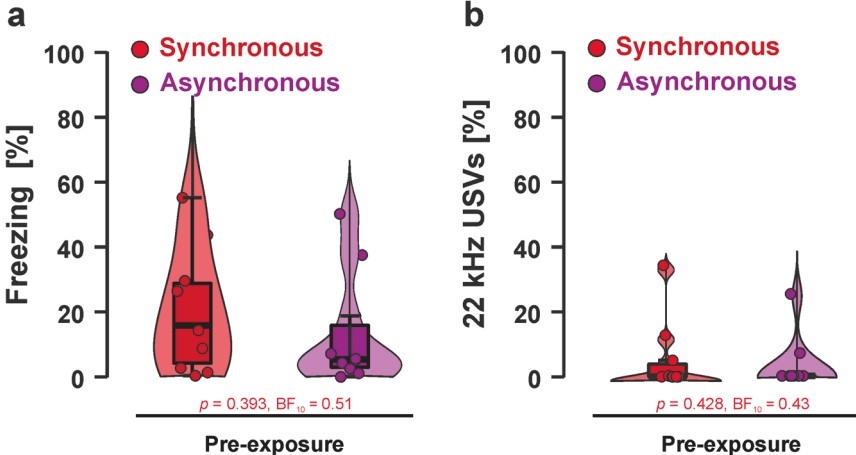

**Fig. 6 Fear responses during pre-exposure.** Neither **a** freezing nor **b** 22 kHz vocalizations differed between the experimental groups during pre-exposure to low amplitude shocks. Boxplots reflect the median, the first and third quartile, and the interquartile range.

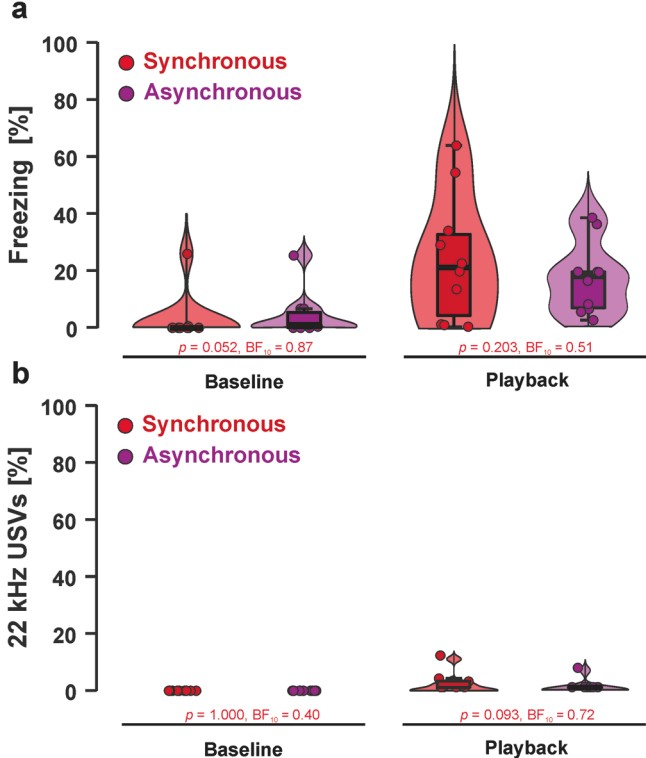

**Fig. 7 Fear responses during playback. a** We found no differences in freezing during baseline or playback between the two experimental groups. Direct comparisons between baseline and playback phases can be found in Supplementary Fig. 12. **b** The same result pattern was found for 22 kHz vocalizations. Direct comparisons between the baseline and playback phases can be found in Supplementary Fig. 13. Boxplots reflect the median, the first and third quartile, and the interquartile range.

statistics for freezing rates and 22 kHz calls during squeak playback are presented in Supplementary Tables 14 and 15, respectively. A time-resolved figure representing freezing responses and 22 kHz calls for each stimulus presentation can be found in Supplementary Fig. 11. Baseline to playback differences can be found in Supplementary Figs. 12 and 13 for freezing and 22 kHz vocalizations, respectively.

## Discussion

The present study investigated the role that pain squeaks have in triggering fear responses (freezing and 22 kHz ultrasonic vocalizations), how the response to these squeaks depends on prior experience with footshocks and whether such pre-exposure to footshocks that trigger squeaks could be substituted by the pairing of a painful stimulus ($CO_2$ laser or shock weak enough not to trigger squeaks) with the sound of pain squeaks, as Hebbian learning or auto-conditioning perspectives may suggest. We focused on pain squeaks, because they had so far not been explored in emotional contagion paradigms, yet occur at a moment that coincides with responses in the cingulate cortex, a region that is necessary for emotional contagion[22].

In the first experiment, we showed that listening to pain squeaks indeed triggered typical fear reactions in rats pre-exposed to footshocks strong enough to elicit squeaks during pre-exposure (0.8 mA), including an increase in freezing and emission of 22 kHz ultrasonic vocalizations. Without pre-exposure, listening to pain squeaks only triggered low freezing rates and no 22 kHz calls. We further found that 22 kHz ultrasonic vocalizations were almost exclusively emitted during intact squeak playbacks and not in

response to phase-scrambled squeaks, suggesting a specificity for squeak vocalizations. Freezing behavior was indistinguishable between animals receiving squeak or phase-scrambled control stimulus playbacks, suggesting that the pre-exposure led to some degree of generalization to sounds resembling pain squeaks in their frequency composition.

In the second experiment, we explored whether the pre-exposure to footshocks, which triggers both an aversive inner state highly effective for fear-conditioning and the emission of pain squeaks and is thus ideal for Hebbian learning and auto-conditioning, could be substituted using a weaker but still painful stimulus (a $CO_2$ laser) paired with squeak playback. We thus employed four experimental groups, in which animals were confronted, during pre-exposure, either with (1) a painful laser stimulation paired with a squeak, (2) only the painful laser, (3) only the squeak, or (4) neither stimuli. During pre-exposure, we observed reliable pain responses (paw or full-body retraction) in the experimental groups receiving high-intensity laser stimulation while these were largely absent from the other conditions. Fear responses upon the playback of pain squeaks 48 h later however were much lower in freezing and absent in 22 kHz calls, with both occurring at significantly lower levels than when animals were pre-exposed to footshocks in Experiment 1. There was strong evidence against a difference between the experimental groups of Experiment 2 during squeak playbacks, preventing any further conclusions with respect to the auto-conditioning hypothesis.

Due to the inconclusiveness of the results from Experiment 2 and the modality change from shock to laser, we conducted a third experiment that procedurally mimicked the experimental design from Experiment 2 but used low-amplitude footshocks instead of a $CO_2$ laser to trigger an unpleasant sensation qualitatively similar to the footshocks of Experiment 1 but low enough in intensity not to systematically trigger squeaks. As in Experiment 2, we then played back squeaks during pre-exposure. Specifically, we synchronized the squeak playback to the shock in our experimental group, to mimic auto-conditioning. In the control group, we still played back the squeaks, but asynchronously, so as to have a similar opportunity for sensitization while perturbing the potential for associative learning central to auto-conditioning. Similar to Experiment 2, there was no observable difference between the two groups. However, fear responses during pre-exposure were also rather mild, calling into question if low amplitude foot shocks triggered an aversive state in the animals sufficient to test the contribution of auto-conditioning during the experience of the more threatening shocks typically used for pre-exposure in the literature and in Experiment 1.

Squeaking as a response to painful electroshocks has been documented for decades in rodents[20] and has recently been mechanistically investigated in mice[42]. Our results from animals that were not pre-exposed to painful experiences (e.g. the NoShock group in Experiment 1 and the Naive group in Experiment 2) show that without additional pre-exposure, squeaks do trigger some nocifensive responses but these are similar in magnitude to those triggered by the playback of phase-scrambled squeaks, and substantially milder than after pre-exposure to squeak inducing shocks. These responses were potentiated by the pre-exposure to the footshocks triggering pain squeaks in Experiment 1. Since the animals in the first experiment did not only show elevated freezing during squeak but also phase-scrambled squeak replay, it is possible that this potentiation could reflect either sensitization to novel stimuli in general or auto-conditioning to squeaks but generalized to similar stimuli such as the phase-scrambled squeaks that shared several features of the original. Given that in auto-conditioning, pain squeaks occur during the US, rather than preceding and predicting the US, ideal conditions for conditioning are not met, and the animals may

have undergone a form of pseudo-conditioning[43], defined as the emergence of conditioned responses in the presence of a CS that was not paired with the US.

The other proxy of fear in the animals, the 22 kHz ultrasonic vocalization showed a more nuanced picture. Such 22 kHz ultrasonic vocalizations are typically emitted in aversive situations, for example, predator exposure, aggressive encounters, or long-lasting social isolation. In the laboratory, they are often seen in response to air puff, acoustic startle stimuli, drug withdrawal, or footshocks[44–46]. However, whether social signals emitted by conspecifics can trigger 22 kHz ultrasonic vocalizations in the receiver and what social signals are particularly efficient in doing so remains unclear. There is anecdotal evidence that the emission of 22 kHz calls by adult rats tested in a visible burrow system elicited 22 kHz calling in offspring[47]. However, under standardized conditions in playback experiments, no prominent induction of 22 kHz ultrasonic vocalizations was found[37,46]. Here, we now show that playback of pain squeaks can trigger a fear response including not only freezing but also 22 kHz calls in the receiver. This finding has important implications as it might help to explain how an aversive experience is shared between conspecifics not directly exposed to the noxious stimulus and how this information is shared in larger social groups living in a wide network of underground tunnels and chambers. Importantly, higher levels of 22 kHz ultrasonic vocalizations were emitted upon hearing the intact than the phase-scrambled squeaks. This highlights the importance of extending our measurements of fear beyond a single behavior (freezing) to better interrogate the internal state of the animals and suggests that rats could indeed have associated somewhat specifically their own squeaks with the shock leading to a conditioned fear recall upon playback of the intact squeaks. It should be noted that there is a possibility that these somewhat specific responses are still due to sensitization as outlined by Parsana et al.[29]: responding to the squeaks could be innate and genetically hard-wired, but freezing and ultrasonic vocalizations may require the animal to have other reasons to be alert and risk aware, for this inborn sensitivity to trigger these observable behaviors—reasons the pre-exposure may have provided.

While 22 kHz vocalizations have been demonstrated to occur during solitude when exposed to predators[47] but see Blanchard et al.[48], during fear learning and subsequent fear recall[49,50] or when for example subjected to aversive handling procedures[44], the difference between freezing and 22 kHz calls in our results may speak to the communicative role of the 22 kHz vocalizations[51]. Although the experiments did not feature another conspecific, the playback of squeaks and the olfactory presence of the bedding smell from other conspecifics likely induced an impression of another rat's presence in the experimental animals since playback took place in the dark. Previous studies have demonstrated that these vocalizations can be specific alarm calls directed at conspecifics to signal potential danger in the environment[15]. It could be speculated that the specificity of the effect could relate to freezing being more prone to sensitization as it is a self-directed response to danger whereas the 22 kHz vocalizations are less prone to sensitization as they are primarily directed towards others. Possibly the intact squeaks may have provided listeners with more reasons to communicate with a conspecific, the presence of which is suggested by the calls, than the phase scrambled squeaks. That is to say, the scrambled squeaks may still be alerting, and trigger freezing, but by being less suggestive of the presence of a conspecific, they may trigger less incentives to emit conspecific directed alarm calls. Future research is however needed to further explore this possibility.

Although the second experiment and the third experiment had the specific aim to shed light on the nature of the process that

accounts for increased vicarious freezing in the literature[9,14,16] and that generated the results from our first experiment, the results remain unfortunately inconclusive. While the results of both experiments speak at first glance against the auto-conditioning and in favor of the sensitization hypothesis, the absence of pronounced fear responses in both experiments when presented with the laser or a low amplitude shock call into question if any fear responses observed during playback were due to sensitization to aversive stimuli at all. In the study of Cruz et al.[17], it was noted that the effects of emotional contagion could only be observed if there was an aversive experience in combination with freezing behavior. Neither the aversive experience nor the freezing by itself was sufficient to induce later emotional contagion upon hearing an interruption of motion sounds. In contrast to our high amplitude footshock pre-exposure, which triggers robust freezing, the laser stimulation and low amplitude footshock did not produce such freezing, and may thus have triggered a state of pain without robust fear. If the animals were to have associated the pain squeaks with this state of pain without fear, the later playback of squeaks would not have triggered freezing or 22 kHz ultrasonic vocalization. Accordingly, our efforts to use a different stimulus ($CO_2$ laser) that does not trigger pain squeaks or a low amplitude footshock that only rarely results in pain squeaks, in order to restrict auto-conditioning to the condition in which we artificially pair the pain with squeaks, appears to have backfired, because both methods failed to create the defensive inner state that manifests in freezing and ultrasonic vocalizations. Another possibility may be that auto-conditioning requires a self-production of the squeak. In Experiment 2 the rats only listened to the playback of a squeak during pre-exposure. While neuronal activity in auditory areas might have been similar, there would have been a lack of motor expression in the laryngeal muscle that is necessary for vocalization[52]. Thus, corresponding motor areas were not activated during laser or low shock pre-exposure. Since affective states are strongly embodied[53,54], the mere perception of the squeak in the absence of any embodiment could have potentially impaired any conditioning of the squeak. In Experiment 3, animals occasionally produced squeaks, but animals that produced more squeaks did not later freeze more or produce more 22 kHz calls in response to squeak playbacks, speaking against the idea that squeak production is key. While the results from the second and third experiments cannot adjudicate directly on the auto-conditioning hypothesis, they still tentatively speak against sensitization as the animals received an aversive painful experience that did not lead to any increase in freezing or 22 kHz calls. Since sensitization effects can also not explain the results on lesioning the auditory thalamus[13], we believe that an interpretation of auto-conditioning to squeaks is more likely to account for our obtained results in Experiment 1.

The present study is subject to limitations that need to be acknowledged. First, the sample sizes in the first experiment are on the low side, especially for the group that did not receive any shocks during pre-exposure. Given the consistency of the data across animals and the support for the effect using Bayes factors, we believe that these results are valid nevertheless. Another limitation of the present study is that we cannot rule out that other cues available for auto-conditioning such as 22 kHz vocalizations might have contributed to the fear responses observed in the first experiment. While the initial fear responses were certainly evoked by the presentation of the playback squeak cues as any baseline period was void of fear responses, the animals started to emit 22 kHz USVs soon after the first playback. As the animals could have auto-conditioned to other fear behaviors during pre-exposure, this could have amplified the fear responses during the playback session. Our study was further limited by the fact that

the laser stimulation did not elicit fear responses that are common for shock delivery. A key difference of our laser stimulation was that its pain was much more focal than the whole-body experience resulting from footshocks. Future studies could use a wider beam or beam splitters to provide a less localized pain sensation or more frequent and intensive laser stimulations below the squeak threshold. This study was also conducted exclusively in male rats. Thus, our results do not necessarily generalize to female rats. Furthermore, female rats show a different behavioral response to fearful stimuli compared to the freezing of males, i.e. darting[55,56] but also in their USV emissions[57]. This darting behavior is constituted by brief and high-velocity movements in the experimental chamber. It could be that differences that were not observed in freezing responses during squeak playback could be detected when analyzing darting behavior. Finally, the cross-experimental comparisons with Han et al.[16] should be treated with a bit of caution since different experimenters were conducting the experiments which could potentially introduce experimenter effects[58].

In conclusion, we could show that pre-exposure to footshocks triggers an increase in emotional contagion to hearing pain squeaks, in line with similar findings for the sound of freezing[17] and 22 kHz calls[20]. Indeed, we found the playback of squeaks suffices to trigger freezing levels in shock pre-exposed animals that were only 20% lower than those triggered by the full experience of witnessing an animal receive the same number of footshocks in similar experiments[16]. We were however unable to find evidence that the pairing of noxious stimuli delivered at intensities that do not trigger pain squeaks paired with hearing squeaks suffices to replicate this potentiation, as auto-conditioning may have suggested. Since there seem to be multiple stimuli that rats can potentially auto-condition to, the well-documented effect of prior experience on the multimodal experience of witnessing other animals in distress in close physical proximity[6] is likely to result from cumulative effects on individual cues. This could explain why the effects of self-experience are more robust in the multimodal real-life situation[8,13,16] than when individual cues are isolated[30]. Taking the results from Experiments 2 and 3 together, it seems unlikely that the null effects from the second experiment were due to changes in modality ($CO_2$ vs. footshocks). Rather, it seems that the painful experience was too low and this apparently prevented forming an association with fear-related cues such as squeaks. Such results are in line with dose-dependent studies on fear conditioning in rats that showed no conditioned fear response for a 0.2 mA shock but reliable conditioned fear responses from 0.5 mA onwards[45]. We tried to increase the potential for fear conditioning of 0.2 mA shocks by increasing the duration to 4 s as the study of Wöhr et al.[45] used 500 ms duration shocks. This however did not seem to increase fear responses during playback. It thus seems likely that testing the auto-conditioning hypothesis for squeaks on a strictly behavioral level faces a conundrum that may very well make the approach impossible: from an experimental point of view, the nociceptive stimulus has to be mild enough not to trigger squeaks systematically, in order to then compare animals with an added squeak playback against those without squeaks, yet, from a threat-conditioning point of view, the noxious stimulus has to be threatening enough, that it actually will produce a squeak. As our results do not negate the auto-conditioning hypothesis, it is important to conduct follow-up studies that go beyond behavioral approaches. For example, future studies could complement our findings using more invasive techniques by for example temporally deafening the animals during shock pre-exposure using pharmacological injections. If the auto-conditioning hypothesis holds true, squeak playback should not induce fear under these conditions. Surgically

interfering with the ability of the animals to produce squeaks is probably not a viable option to test auto-conditioning to audible pain-squeaks. Indeed, when Calub et al.[30] devocalized rats through a unilateral transection of the recurrent laryngeal nerve to test auto-conditioning for 22 kHz calls, this reduced the rats' emission of 22 kHz calls during shock pre-exposure. Specifically, of the 15 sham-operated animals, 12 emitted 22 kHz calls during shock pre-exposure while of the 16 animals having undergone the devocalization surgery, only 5 did. Using a chi-square test, this is a significant reduction ($\chi^2 = 7.42$, $p = 0.006$, $BF_{10} = 15.69$). The same was not true for the pain-squeaks: although statistics on the audible pain squeaks were not reported in the paper, upon our request, author Sharon Furtak re-examined the audio recordings during shock administration, and reported that of the 15 sham animals, 11 emitted audible squeaks, and of the 16 animals having undergoing the devocalization surgery, 12 still emitted audible pain squeaks (Sharon Furtak, personal communication). Using a chi-square test, this finding leans towards evidence against an effect of this surgery on the emission of audible squeaks ($\chi^2 = 0.011$, $p = 0.916$, $BF_{10} = 0.374$). Furthermore, it would be interesting to investigate the neurobiological difference between animals with and without self-experience during squeaking in emotional contagion either by playback or using a demonstrator behind an opaque divider. Here, areas such as the insula or ACC would be of particular interest due to their known contribution to emotional contagion and pain mirror responses in rodents and humans[3,22,59–61]. Finally, it could be interesting to investigate the role of familiarity with the emitter of the squeaks as the present study played back squeaks of unknown rats. Previous research in mice has demonstrated that familiarity during an observation of shock increases the transmission of fear[62]. However, our own experiments have shown that familiarity does not play a similar role in rats: rats show fear responses even when witnessing unfamiliar rats receive shocks, and higher levels of familiarity do not translate into robustly higher fear responses in observers[16]. Therefore, such a follow-up seems more feasible in mice.

## Methods

**Subjects.** Twenty-four adult male Long Evans rats (6–8 weeks old; 250–350 g; Janvier, France) were used as experimental subjects in the first experiment. The animals were randomly assigned to the experimental groups upon arrival in the local animal facility at the Netherlands Institute for Neuroscience where animals were housed socially (Type IV cages with two to four animals per cage) with ad libitum access to food and water in a specific pathogen-free room controlled for temperature (22–24 °C), relative humidity (55%), and lighting (12 h reversed light/dark cycle). All experimental procedures were approved by the Centrale Commissie Dierproeven (CCD number: AVD801002015105) and by the welfare body at the Netherlands Institute for Neuroscience (study dossier number: NIN181109). The experiment was carried out complying with all ethical regulations regarding animal testing.

Eighty adult male Long Evans rats (6–8 weeks old; 250–350 g; Janvier, France) were used as experimental subjects in the second experiment. As for Experiment 1, the animals were randomly assigned to the experimental groups upon arrival at the local animal facility. Housing conditions were identical to Experiment 1. All experimental procedures were approved by the Centrale Commissie Dierproeven (CCD numbers: AVD801002015105 and AVD8010020209724) and by the welfare body at the Netherlands Institute for Neuroscience (study dossier numbers: NIN201101 and NIN203701). The experiment was carried out complying with all ethical regulations regarding animal testing.

Twenty adult male Long Evans rats (6–8 weeks old; 250–350 g; Janvier, France) were used as experimental subjects in the third experiment. As before, the animals were randomly assigned to the experimental groups upon arrival at the local animal facility. Housing conditions were identical to Experiments 1 and 2. All experimental procedures were approved by the Centrale Commissie Dierproeven (CCD number: AVD8010020209724) and by the welfare body at the Netherlands Institute for Neuroscience (study dossier numbers: NIN223708). The experiment was carried out complying with all ethical regulations regarding animal testing.

**Experimental groups**. For Experiment 1, animals were divided into three experimental groups based on the pre-exposure condition and the test stimuli to which they were subjected. Animals in two of these groups were pre-exposed to footshocks prior to the auditory playback tests. During the tests, one of these groups was presented with previously recorded squeak vocalizations (Shock → Squeak group, $n = 10$), whereas the other group was presented with control stimuli synthesized from the original squeaks (Shock → Control group, $n = 9$, see the subsection "Stimuli" for details). Animals in the third group were not pre-exposed to footshocks and were tested with the original squeaks (NoShock → Squeak group, $n = 5$).

For Experiment 2, animals were divided into four experimental groups based on the pre-exposure condition to which they were subjected (each $n = 20$). Animals in one of the groups were only exposed to a painful $CO_2$ laser stimulation during pre-exposure (Laser group), whereas the animals in another group were administered the same levels of laser simultaneously with auditory playbacks of previously recorded squeaks (Laser + Squeak group). Animals in another group received only squeak playbacks (Squeak group), while the animals in the last group were neither subjected to squeak playbacks nor painful levels of the laser (Naïve group).

For Experiment 3, we divided the animals into two experimental groups based on different pre-exposure conditions (each $n = 10$). Animals in the first group received low-amplitude shocks simultaneous to the presentation of squeak playbacks similar to the Laser + Squeak group in Experiment 2. This group was thus labeled as the Synchronous group. In the second group, animals also received low-amplitude shock but the squeak playback was delayed into the ITI. Thus, the group was labeled as the Asynchronous group.

**Stimuli**. Original squeak vocalizations were recorded from adult male Long-Evans rats receiving footshocks (1 s, 1.5 mA) during an emotional contagion test published elsewhere[16] using a CM16/CMPA condenser ultrasound microphone with an Ultra-SoundGate 116Hn audio recording system and the Avisoft-RECORDER software (Avisoft Bioacoustics, Germany). Five different squeak exemplars were recorded from each of the three different rats for generalizability. These recordings were manually trimmed, tapered with a Tukey window, and root-mean-square-amplitude-normalized over the entire duration, resulting in 15 individual squeaks with a 1.077 s mean duration (SD = ±0.096 s). The control stimuli were synthesized from the original squeaks via Fourier-transforming the signal first, then randomly shuffling the phase spectrum, and finally inverse Fourier-transforming the signal. This procedure ensures that the temporal structure of the sound is entirely taken out, while the spectral structure over the whole duration of the sound remains intact. We chose to use these phase-scrambled squeaks as control stimuli because our pilot studies suggested that their playback elicited reduced levels of freezing.

For Experiment 1, any experimental rat was only presented with the original or the phase-scrambled versions of the five squeaks of only one of the three rats from which the squeaks were recorded. An example squeak, together with the corresponding phase-scrambled control version, is shown in Fig. 8a and b (spectrogram) and 8c and d (normalized amplitude), respectively. All squeaks, the underlying raw data, and the analysis code can be found at https://osf.io/efuq4/[63].

The sound pressure levels of the original squeaks were measured to be 90 dB on average (range: 88–92 dB) by a microphone located above the center point of the observer chamber in the emotional contagion setup described in Han et al.[16]. These sound pressure levels were quantified using the

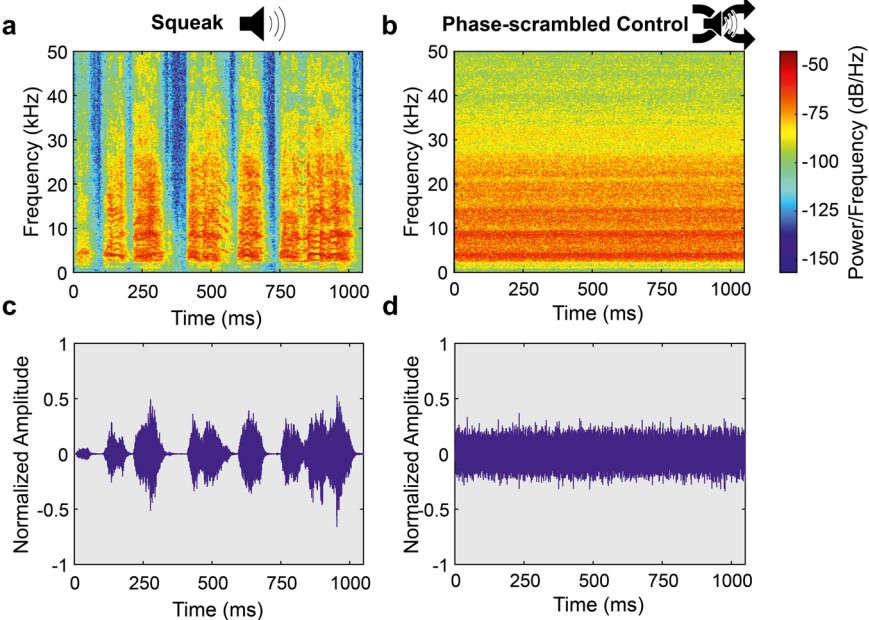

**Fig. 8 Employed stimulus material.** Spectrogram of a regular squeak (**a**) and a phase-scrambled squeak (**b**). Normalized amplitude of a regular squeak (**c**) and a phase-scrambled squeak (**d**).

**Table 1 Number of animals per group with respect to different amplitudes during playback.**

| | Playback at 75 dB | Playback at 85 dB | Freezing (75 vs. 85 dB) | 22 kHz USVs (75 vs. 85 dB) |
|---|---|---|---|---|
| Shock → Squeak | $n = 5$ | $n = 5$ | $t(8) = 0.50$ $p = 0.631$ $BF_{10} = 0.532$ | $t(8) = 0.92$ $p = 0.383$ $BF_{10} = 0.637$ |
| Shock → Control | $n = 5$ | $n = 4$ | $t(7) = 1.56$ $p = 0.162$ $BF_{10} = 0.981$ | $t(7) = 1.40$ $p = 0.204$ $BF_{10} = 0.875$ |
| NoShock → Squeak | $n = 0$ | $n = 5$ | | |

Calibrated 40 kHz Reference Signal Generator in combination with the Avisoft-SASLabPro software version 5.2.13 (Avisoft Bioacoustics, Germany). Specifically, the sound pressure level of each squeak was measured by first automatically segmenting the individual bouts that make up the squeak, then calculating the root-mean-square-amplitude of each bout in dB, and finally taking the average dB of all the bouts. In the current study, all stimuli were played back at either 75 or 85 dB. This difference was initially used to investigate the possible effects of amplitude on animals' fear responses. In total, five animals from the Shock → Squeak and the Shock → Control group and no animal from the NoShock → Squeak group received squeak playbacks at 75 dB. Five animals from the Shock → Squeak group, four animals from the Shock → Control group, and five animals from the NoShock → Squeak group were presented with stimuli at 85 dB (see Table 1 for a summary). We decided to pool the results of these two amplitude levels as they led to highly comparable patterns of fear responses. To account for this experimental difference, we included the amplitude of the playback as a covariate in the statistical model (see the subsection "Statistics and reproducibility" for details).

Stimuli used in Experiment 2 during pre-exposure and auditory playback tests were identical to the original squeaks and phase-scrambled control sounds described in Experiment 1. For animals that were presented with squeak playbacks during pre-exposure, different squeaks were used during the playback tests. In Experiment 2, all stimuli were played back at 75 dB.

Auditory stimuli used in Experiment 3 during pre-exposure and auditory playback tests were identical to the original squeaks described in Experiment 1. However, as the shocks during pre-exposure in Experiment 3 were 4 s long instead of the 1 s of Experiment 1, we combined four 1 s recordings into a 4 s audio stimulus. Scrambled squeaks were not included as we could not show any innate aversive response to scrambled squeak playbacks in Experiment 1. For the playback period, we used the 1 s long squeaks as in Experiments 1 and 2 to ensure comparability of fear responses between experiments. Identical to Experiment 2, all stimuli were played back at 75 dB.

**Apparatus**. In Experiment 1, pre-exposure with footshocks was delivered in a custom-built pre-exposure chamber ($L$: 30 cm × $W$: 20 cm × $H$: 40 cm) featuring two experimental chambers divided by a transparent perforated separator. As contextual markers, the walls of the pre-exposure chamber were covered with black and white stripes, the overhead daylights were turned on, the background radio was turned off, and the chamber was wiped with a vanilla aroma after cleaning with rose-scented dishwashing soap. During the experimental procedure, the animals were placed on stainless steel grid rods of one of the experimental chambers through which electrical currents could be applied to the animals via a stimulus scrambler (ENV 414-S, Med Associates Inc., VT). USVs were only recorded for the 10 animals receiving lower amplitude (75 dB) playback. Auditory playbacks were administered in another room in a different test chamber ($L$: 24 cm × $W$:

25 cm × $H$: 34 cm) consisting of two adjacent compartments with a transparent perforated divider in between. This test context differed from the pre-exposure context to avoid contextual fear conditioning: the walls of the testing chamber were made of transparent Plexiglass, the lights were turned off, the background radio was turned on at low levels, and a lemon-scented dish-washing soap was applied to the chamber after cleaning with 70% ethanol. Behavior was recorded using a Basler GigE camera (acA 1300-60g), which was mounted to the ceiling of the test chamber and controlled by EthoVisionXT (Noldus, the Netherlands). The ultrasound microphone described above was positioned on top of the compartment that the animal was in, while a Vifa ultrasonic dynamic speaker (Avisoft Bioacoustics, Germany) was positioned in the adjacent compartment facing the animal's compartment. During both the habituation and the auditory playback test phases, bedding material from an unknown, unstressed male rat was placed in the adjacent chamber to prime the rats to the possibility that there was another rat in the vicinity.

Pre-exposure for Experiment 2 was performed in a custom-built pre-exposure chamber different from the one used in Experiment 1 ($L$: 30 cm × $W$: 15 cm × $H$: 30 cm) and placed in a different room inside a Faraday cage. It consisted of a rectangular dark-colored apparatus, opened along one of the long sides with a 0.5 cm fence. The opened side was placed in front of an opening in the Faraday cage. The $CO_2$ laser (CL15 model: M3) was placed outside the Faraday cage and the arm used for delivering the heat pulses protruded into the Faraday cage, with its tip 15 cm away from the observer's box. As in Experiment 1, pre-exposure took place under normal light conditions, the background radio was turned off, and the apparatus was wiped with a vanilla aroma after cleaning with a rose-scented dishwashing soap. Auditory playback tests took place in the same test chamber and room as in Experiment 1. Again, contextual differences were maximized by performing the auditory playback tests in dark conditions, turning on the radio at low volume, and applying a lemon-scent before the test session. As described for Experiment 1, the ultrasound microphone and the speaker were placed in the adjacent empty chamber for recording and stimulus playback, together with bedding material from an unknown male rat.

Experiment 3 used the same chambers for pre-exposure and playback as Experiment 1. All contextual manipulations of the apparatus across experimental days were kept the same between these two experiments to allow for maximal comparability. We also had an ultrasound microphone in the pre-exposure chamber allowing for the recording of 22 kHz vocalizations in addition to identifying how often the animals squeaked.

**Experimental procedure**. For Experiment 1, after acclimatization to the local animal facilities for at least one week, the animals were handled for 5 min each day for 3 days. Then the experimental procedure started (Fig. 9a). On Day 1, all rats were habituated to the auditory playback test context by allowing them to freely explore the chamber for 20 min in the dark. At the end of habituation, the animals were taken out of the chamber and

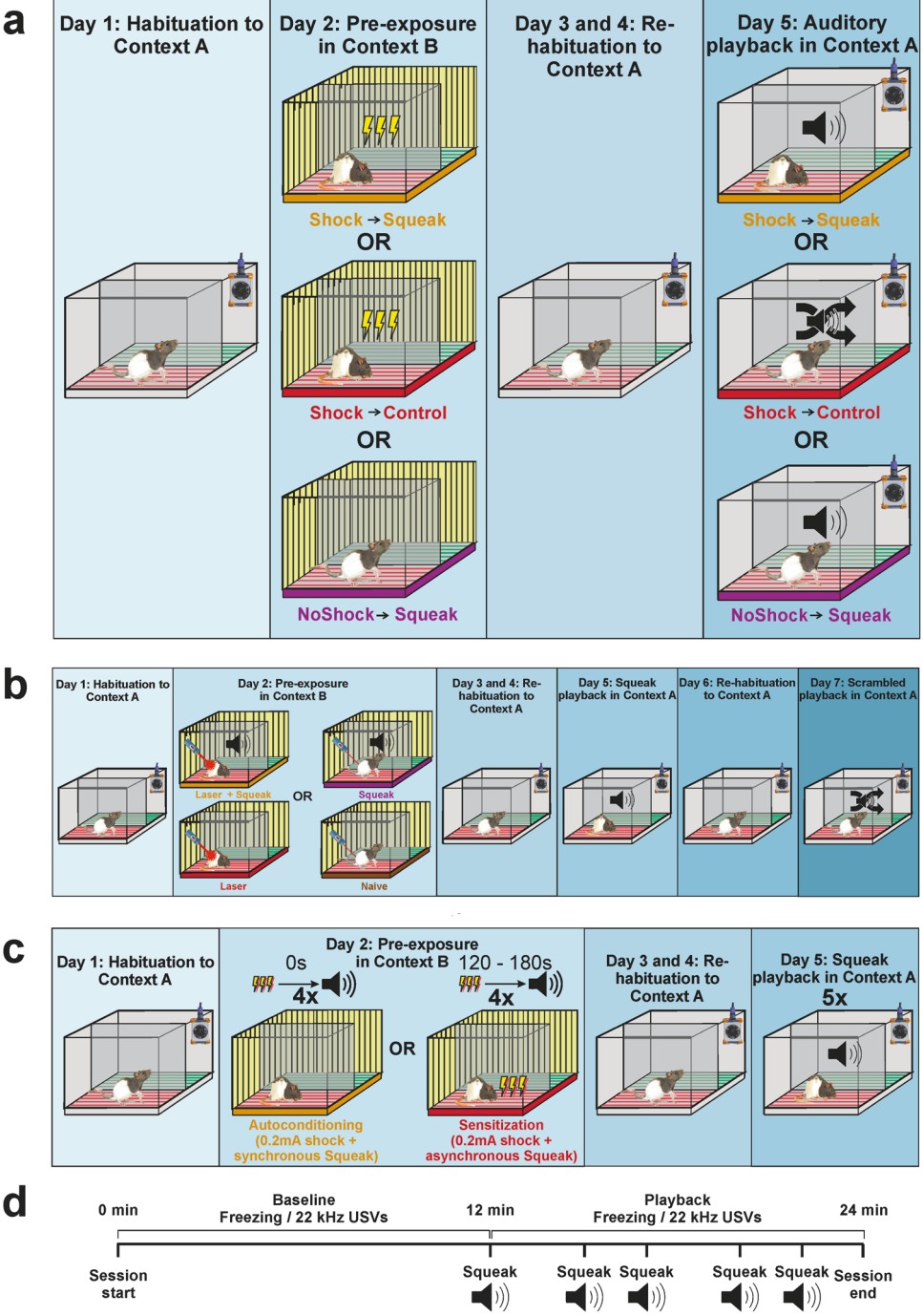

handled for 5 min. On Day 2, the animals underwent the shock pre-exposure procedure in the pre-exposure context in light conditions. The Shock → Squeak and the Shock → Control groups received four unpredictable shocks (1 s, 0.8 mA) with an intershock interval of 240–360 s. The animals in the NoShock → Squeak group were also placed in the same chamber for the same amount of time but did not receive any shocks. At the end of pre-exposure, all animals were first handled for 5 min and then rested individually for one hour in a transportation cage to prevent negative emotional contagion in the home cage. After the pre-exposure day, there were two more habituation days in the playback test context (Day 3 and 4 in Fig. 2a) administered exactly as in Day 1 to reduce baseline freezing and fear in the later test session.

On Day 5 the auditory playback tests were administered. After a 12-min baseline, five playbacks of either the original squeaks or the phase-scrambled control stimuli were administered with an interstimulus interval of 120–180 s (Fig. 9d). The timing and number of playbacks were chosen to match that of our experiments involving shocks to a demonstrator in the neighboring compartment[16,56], to enable comparison of freezing levels across experiments. Animals stayed in the test chamber for another 120 s after the presentation of the last stimuli. The stimuli were played back using the playback system described above with the Avisoft-RECORDER software at a sampling rate of 250 kHz. The sound amplitude levels were calibrated and adjusted with the calibrated 40 kHz Reference signal generator in combination with the Avisoft-SASLabPro software version 5.2.13. At the end of the

**Fig. 9 Experimental paradigm and timeline. a** Experiments took place in a two-compartment chamber separated by a transparent divider. Pre-exposure on day 2 could either consist of footshocks (Shock → Squeak and Shock → Control groups) or a resting period (NoShock → Squeak group). Playback on day 5 was of regular squeaks (Shock → Squeak and NoShock → Squeak groups) or phase-scrambled squeaks (Shock → Control). Note that the yellow background in the pre-exposure box symbolizes the fact that overhead lights were turned on in context B but not A. **b** Behavioral paradigm of the second experiment. The procedure only differed between groups on the day during pre-exposure. Here, groups either received laser stimulation together with squeak playback (Laser + Squeak), only laser stimulation (Laser), only squeak playback with a low-intensity laser stimulation (Squeak), or a low-intensity laser stimulation only (Naïve). On day 5, regular squeaks were played back to all animals whereas phase-scrambled squeaks were played back on day 7. Note that the yellow background in the pre-exposure box represents the turned-on overhead lights. Timing of the experiment mimicked the schedule of Experiment 1. **c** Experimental paradigm of Experiment 3. Experimental procedures on days 1, 3, 4, and 5 were kept identical to Experiment 1. On day 2 however, we presented all animals with low amplitude shocks (0.2 mA, 4 s; 0.2 mA was chosen in a pilot study as a threshold intensity that triggered squeaks in only rare instances) that were either synchronously paired with 4 s of pain squeak playback that covered the entire period of shock delivery or asynchronously, with the 4 s squeak playback presented exactly in the middle of the 240–360 s interval between two shocks. **d** Experimental timeline for the auditory playback session on day 5. Baseline freezing and 22 kHz vocalizations were measured across the entire 12 min baseline period. Playback freezing and 22 kHz vocalizations were computed across the entire 12 min playback period. The time-resolved stimulus-by-stimulus presentations of fear responses use the intertrial interval (ITI) between individual squeaks for quantification. Note that ITIs between squeaks were randomly chosen to be either 120 or 180 s. The depicted sequence is thus an example of a possible randomization order.

auditory playback tests, animals were housed individually for one hour to prevent stress contagion in the home cage.

In Experiment 2, all animals were acclimatized to the local animal facility and handled as in Experiment 1. Animals were habituated for 10 min to the auditory playback test context on the first experimental day and for two consecutive days after pre-exposure (see Fig. 9b). For the first 40 animals (each experimental group containing ten animals) tested in this experiment, no habituation to the pre-exposure context was provided. However, due to elevated freezing responses in all experimental groups during the pre-exposure session (even in the Naïve group), the remaining 40 animals received one session of habituation in the pre-exposure context one day before the pre-exposure procedure. Furthermore, the laser was applied for this subset of animals through a hole in the closed door preventing any visual contact of the animal with the experimenter. While both these differences in the procedure should have minimal impact on the auditory playback session, they might have affected the fear responses during the pre-exposure session. Thus, habituation (absent for the first 40 and present for the last 40), was included as a covariate in the statistical model for pre-exposure analyses (see the subsection "Statistics and reproducibility" for details).

During pre-exposure, the animals could freely explore the arena to prevent any restraining stress. The $CO_2$ laser was applied manually by the experimenter to the hind paw of the animal at an approximate distance of 10–15 cm. For all groups, pre-exposure started with a 2-min baseline period, in which the animals were administered six below-threshold (30% of laser power) laser stimulations (200 ms) to habituate them to the laser arm entering the chamber and the click sound associated with laser delivery. For the Laser group, four laser stimulations at 70% intensity of the laser power to one of the paws of the animal were then administered with an intertrial interval of 240–360 s. This stimulation level of the $CO_2$ laser was previously shown to be effective in eliciting pain[22]. For the Laser + Squeak group, the same procedure was applied with the addition of the playback of a previously recorded squeak that started at the same time as each laser stimulation. Here, the laser stimulation triggered squeak replay with close to zero latency to simulate a situation similar to shock experience, in which pain squeaks are measured with short latency following the onset of footshocks[22] (Fig. 4a). For the Squeak group, the laser intensity was reduced to the below-threshold level, which does not elicit pain (30% of laser power), and squeaks were played back as described above. Finally, for the Naive group, laser intensity was again reduced to the below-threshold levels and no squeaks were played back upon stimulation. Low-intensity laser was used instead of no laser

stimulation alone, to control for the aiming of the laser onto the animal and the faint clicking associated with delivering the laser. After the last stimulus delivery, the animals were left for 4 min in the apparatus before being handled for 5 min and finally rested individually in a separate room in a transportation cage for two hours to prevent stress contagion in the home cage that they shared with two or three other rats (depending on the batch size) of the same experimental condition.

Auditory playback tests were performed exactly as described for Experiment 1 except that on day 5 following the third habituation, all groups were first tested with squeak playbacks. On day 6, all animals were reintroduced to the test chamber and exposed to the context for 10 min to extinguish any potential negative association that might have formed due to squeak playbacks. On day 7, all groups were this time tested with phase-scrambled control sounds. Thus, in contrast to Experiment 1, responses to squeaks and control sounds were tested serially in a within-subject, rather than in a between-subject design.

Acclimatization and handling in Experiment 3 were conducted as in Experiments 1 and 2. On Day 1, all rats were habituated to the auditory playback test context in the dark for 20 min. The animals were then taken out of the chamber and handled for five minutes. On Day 2, the animals underwent the shock pre-exposure procedure in the pre-exposure context in light conditions. The Synchronous and the Asynchronous groups received four unpredictable shocks (4 s, 0.2 mA) with an intershock interval of 240 or 360 s. The intensity of 0.2 mA was chosen based on a previous study that had shown that animals show immobility/freezing when presented with shocks at this amplitude compared to a no-shock control group[45] suggesting that they are triggering fearful states in rats. Importantly, in a pilot study using four animals, we tried different shock intensities and found that using higher intensities (0.3 mA) already triggered squeaks invariantly, making 0.2 mA the highest intensity suitable to dissociate the shock from hearing oneself squeak. The animals in the Synchronous group were played back 4 s recordings of pain squeaks during the shock presentation. To match the duration of the shock with the squeak playback, the squeaks were played back for the entire 4 s duration. Animals in the Asynchronous group received the playback exactly in the middle of the interval between two shocks. Thus, the time difference between shock presentations and the following squeak playback was always either 120 or 180 s (average 150 s across trials), and the same was true between the interval between a shock and the preceding playback. These long intervals should minimize both forward and backward conditioning. All procedures following pre-exposure were identical to Experiment 1 and the full experimental procedure is detailed in Fig. 9c.

**Freezing and 22 kHz call quantification.** Freezing for all Experiments was scored in BORIS v 7.7.3[64] by an experimenter blind to the experimental conditions. During the auditory playback tests, an infrared LED was attached to the setup to provide visual feedback in the video recordings when a stimulus was presented. A threshold of minimally 3 s of freezing was used to ensure that the scored behavior was actual freezing and not just transient immobility. Percentages of time spent freezing within the baseline and the auditory playback periods (12 min each) were calculated separately for statistical analyses. 22 kHz vocalizations were semi-automatically detected using the MATLAB toolbox DeepSqueak (version 2.5.0, Long Rat Call_V2 network with default settings[65]) first, and then manually checked by an operator blind to the experimental conditions. Identically to the freezing scores, percentages of time spent emitting 22 kHz calls within the baseline and the auditory playback periods were calculated separately for statistical analyses. Opposed to experiments involving a real demonstrator where attributing vocalizations to the observer or demonstrator can be difficult, all vocalizations in our experiment (except those involved in the playbacks) could unambiguously be attributed to the observer. Freezing and 22 kHz vocalizations during the auditory playback sessions were quantified during the baseline period (percentage of time spent freezing or vocalizing relative to the total baseline period length, i.e. 12 min) as well as for the playback period (percentage of time spent freezing or vocalizing relative to the playback period length). Since both periods were always 12 min long, a freezing value of 100% would thus indicate that animals were freezing for the entire 12 min in the respective experimental period. In Experiment 1, squeaks during pre-exposure were identified directly from the audio channels of the video recordings, as USV microphones were not present for all animals during this period.

For the second experiment, we first explored the behavioral responses during pre-exposure to validate that pain was reliably induced via laser delivery as the application of a $CO_2$ laser as an aversive tool is less common[40,41]. We calculated a custom pain score across all four laser stimulations based on the behavioral responses of the animals following stimulus presentation. Here, animals received a score of 0 if they did not react at all, a score of 1 if they slightly twitched without retracting the limb, a score of 2 if they retracted the targeted limb, and a score of 3 if they retracted the limb and moved from their current location. Thus, the total pain score could vary between 0 (minimum) and 12 (maximum). Quantification of freezing and 22 kHz calls were performed exactly as described above, except that an additional side view camera in addition to the top view camera, was available for scoring freezing in the auditory playback tests.

Quantification of freezing and 22 kHz vocalizations in Experiment 3 was done identically as in Experiments 1 and 2. We also assessed the number of pain squeaks per shock in all animals in the Asynchronous group. This was not possible in the Synchronous group due to the playback of the pain squeak at the time of shock.

**Statistics and reproducibility.** Statistical analyses were performed using R (version 4.1.3) and JASP (version 0.16.1). For Experiment 1, differences in freezing responses between the experimental groups during pre-exposure were assessed using a one-way ANOVA or Kruskal–Wallis test if normality was violated as measured via a Shapiro–Wilk (abbreviated as S–W) test. For 22 kHz vocalizations, we used an independent sample t-test or its non-parametric counterpart since we only recorded data from two groups (Shock → Control and Shock → Squeak). For the playback session, increases from baseline to playback period for each experimental group were tested by applying either paired

t-tests or Wilcoxon signed-rank tests depending on normality violations. To determine differences between the groups, the baseline and playback period were analyzed separately using a one-way ANOVA (levels: Shock → Squeak, Shock → Control, NoShock → Squeak) if normality assumptions for the residuals were met. For the playback session, the amplitude of the squeak (75 vs. 85 dB) was included as a covariate to account for the potential influence of the loudness of the auditory playback. If normality was violated (S–W p-value < .05), we instead calculated a non-parametric Kruskal–Wallis test to identify differences between the experimental groups. In an additional analysis, we also calculated either a one-way ANOVA or a Kruskal–Wallis test for the difference score between the baseline and auditory playback period to control for potential variability in baseline fear responses. This is similar to assessing the interaction effect of Epoch (Baseline vs. Playback) × Group (Shock → Squeak, Shock → Control, NoShock → Squeak), but because there is no well-established non-parametric test to examine such interactions, assessing the effect of group on the Playback-Baseline measures seemed a more robust alternative. Again, amplitude was included in the model if parametric tests could be applied. Significant main effects were investigated post hoc using parametric t-tests or non-parametric Wilcoxon rank-sum tests. All tests were conducted two-tailed. Because only two of the potential three pairwise comparisons have meaning, we used planned comparisons to assess the effect of pre-exposure (Shock → Squeak vs. NoShock → Squeak) or the specificity for squeaks (Shock → Squeak vs. Shock → Control) that do not require correction for multiple comparisons[66].

The same factorial designs were also analyzed with Bayesian ANOVAs, which have an advantage over frequentist statistics as they can quantify not only the evidence for the presence of an effect but also the evidence for the absence of an effect, as well as the absence of evidence for either[6]. For main effects and interactions, the $BF_{incl}$ was used as a marker of evidence. For post hoc tests of main effects, the $BF_{10}$ was used. For both the $BF_{incl}$ and the $BF_{10}$, a value of >3 is considered to provide evidence in favor of the alternative hypothesis, whereas a value of <1/3 provides evidence in favor of the null hypothesis. Interpretation of the Bayes factors followed the guidelines by Lee and Wagenmakers[67]. All analyses were performed with default prior settings in JASP and effects were estimated across all models.

Since we measured freezing for all animals as well as 22 kHz vocalizations for 10 animals during pre-exposure, we used this as a predictor for the freezing response or 22 kHz vocalizations during playback to identify whether a stronger aversive experience during shock pre-exposure also leads to higher fear response during playback. In addition, we correlated freezing responses and 22 kHz vocalizations during pre-exposure and playback to identify if these measures are associated with each other. As for the ANOVAs, we used both frequentist and Bayesian regression/correlation to provide a comprehensive overview.

We also investigated shock-induced changes during the pre-exposure session in more detail by analyzing the rapid temporal dynamics of 22 kHz call emission, squeak production, and freezing behavior from 5 s prior to 10 s after the shock onset. For each of these time periods, if more than 0.5 s of a 1 s bin contained the behavior under investigation, that bin was scored as 1, if not, scored as 0. Proportions for each bin were then calculated separately by pooling all 4 shock trials of all animals that were analyzed for that particular behavior. For 22 kHz calls and squeaks, this analysis was based on a total of 10 animals (five later on tested with low amplitude squeaks, and five with low amplitude control stimuli), since we only had ultrasound

recordings in these animals during pre-exposure. For freezing, all 24 animals were used. Note that since four shocks are given per animal, the period from −5 to 0 s is not a true baseline, but an interval that in 3/4 of cases occurs after another shock (the inter-shock interval of either 240 or 360 s). Statistical significance of observed proportions was analyzed by comparing each of the observed proportions after shock onset to the average of the proportions observed during the 5 s baseline period via separate binomial tests. The significance level was Bonferroni-corrected at alpha = 0.05/10 for each set of analyses.

For Experiment 2, we first determined whether the experimental groups differed in terms of pain scores, freezing, or 22 kHz calls during pre-exposure using a parametric one-way (Bayesian) ANOVA or a non-parametric Kruskal–Wallis test with the between-subjects factor Group (Laser + Squeak, Laser, Squeak, Naive). Habituation to the pre-exposure context (Yes, No) was included as a covariate if normality assumptions were not violated. Video data (pain and freezing) for two animals in the Naïve, two animals in the Laser + Squeak and one animal in the Squeak condition could not be evaluated due to technical issues in the pre-exposure session.

For the auditory playback sessions using squeaks (day 5) and control sounds (day 7), we repeated the analysis procedure from Experiment 1. First, increases from baseline to playback period for each experimental group were tested by applying either paired t-tests or Wilcoxon signed-rank tests. Then, we computed one-way (Bayesian) ANOVAs or Kruskal–Wallis tests with the between-subjects factor Group (Laser + Squeak, Laser, Squeak, Naive) for the baseline period using both freezing and 22 kHz calls as the dependent variable. This was then repeated for the data from the playback period and for a difference score subtracting the baseline from the playback data. Analysis for the squeak sound playback (day 5) and control sound playback (day 7) were analyzed separately.

As in Experiment 1, we also used freezing responses during pre-exposure as a predictor for freezing during playback. Since we also recorded 22 kHz vocalizations during pre-exposure in this experiment, the same procedure was applied for this measure of discomfort. Finally, we correlated freezing responses and 22 kHz vocalizations during playback to identify if these measures are associated with one another.

Audio data from three animals (one animal from the Laser + Squeak, Laser, and Naïve groups each) from the squeak playback session and six animals (one animal from the Laser + Squeak and Naïve groups each and two animals from the Laser and Squeak groups each) for the control playback session could not be evaluated due to technical issues.

In Experiment 3, we used independent samples t-test or Mann–Whitney U tests if parametric requirements were violated to determine differences in freezing and 22 kHz vocalization between groups during pre-exposure. Freezing responses and 22 kHz vocalizations were compared between the Synchronous and Asynchronous groups using the same tests for the playback period on day 5. To account for potential baseline differences, we again computed a difference score between playback and baseline. Differences from baseline to playback within each group were tested using dependent sample t-test or Wilcoxon signed-rank tests if parametric requirements were violated. We also used the fear responses during pre-exposure as a predictor for the fear responses during playback to identify if rats with stronger fear responses during pre-exposure would react more strongly during pain squeak playback. Finally, we correlated freezing responses and 22 kHz vocalizations during pre-exposure and playback to identify if these measures are associated with one another.

**Reporting summary.** Further information on research design is available in the Nature Portfolio Reporting Summary linked to this article.

## Data availability
All data are fully available under the following link:https://osf.io/efuq4/[63].

## Code availability
All codes are fully available under the following link:https://osf.io/efuq4/[63].

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

## Acknowledgements

This work was supported by the German National Academy of Sciences Leopoldina (LPDS 2021-05; J.P.), the Dutch Research Council (VICI 453-15-009; C.K.), the Fonds Wetenschappelijk Onderzoek—Vlaanderen (FWO; Research Foundation—Flanders; e.g. G0C0522N and G0E6722N; M.W.) and a start-up grant at KU Leuven (PXF-E0120-STG/20/062; M.W.). We thank Sharon Furtak for reanalyzing the audio recordings of the Calub et al. (2018) study to examine whether devocalization affected the emission of pain squeaks.

## Author contributions

Conceptualization: J.P., E.P., E.S., F.M., M.W., V.G., C.K. Methodology: J.P., E.P., E.S., F.M., V.G., C.K. Software: J.P., E.P., E.S., F.M. Formal analysis: J.P., E.P., E.S., F.M., V.G., C.K. Investigation: J.P., E.S., E.P., F.M., E.R., N.S., S.M. Resources: E.S., M.W. Data curation: J.P., E.S., V.G., C.K. Writing—original draft: J.P. Writing—review and editing: E.S., E.P., F.M., E.R., N.S., S.M., M.W., V.G., C.K. Visualization: J.P., V.G., C.K. Super-vision: V.G., C.K. Project Administration: V.G., C.K. Funding acquisition: M.W., V.G., C.K.

## Competing interests

The authors declare no competing interests.
