## [Peer review file · Communications Biology]

Reviewers' comments:

Reviewer #1 (Remarks to the Author):

The authors tested a specific hypothesis that 'auditory auto-conditioning' in observer (subject) rats contributes to emotional contagion phenomenon. To test this, study 1 examined three groups: (i) footshock pre-exposed rats tested with auditory playback of squeak (Shock->Squeak), (ii) footshock pre-exposed rats tested with auditory playback of scrambled squeak (Shock->Control), and (iii) no footshock pre-exposed rats tested with auditory playback of squeak (Naïve->Shock). The main findings were that during the auditory playback testing, both Shock->Squeak and Shock->Control groups showed comparably enhanced freezing compared to the Naïve->Shock group but only the Shock->Squeak animals emitted significant 22 kHz USVs. Study 2 then tested fear responses to squeak playback in (i) rats pre-exposed to a CO2 laser-induced pain paired with squeak playback (Laser+Squeak), (ii) CO2 laser-induced pain alone (Laser), (iii) squeak playback alone (Squeak), and (iii) neither CO2 laser nor squeak playback (Naïve) and found no significant group differences. Based on these results, the authors concluded that "We thus demonstrated the sufficiency of pain squeaks to trigger fear in conspecifics in ways that depend on prior shock experience..."

In my view, this is a very specialized study that yielded potentially interesting findings for a narrow audience. However, the study lacks the following crucial details:

1. Study 1 needs to show freezing, 22-kHz USV and squeak responses in conjunction with four footshocks for two pre-exposed groups. This is necessary to demonstrate the temporal pairing of footshocks with animals' fear responses.
2. The Pain squeak is defined as those in the audible range (line 91). Fig. 1, however, shows that the squeak spectrogram spans 0-30 kHz and thus includes USV. More information is needed on how squeaks were recorded in relation to footshocks and inter-shock intervals. Fig. 1 only shows 1-sec epoch, which is not too informative.
3. Related to comment 2, a crucial control group, i.e., footshock pre-exposed rats tested with auditory playback of 22-kHz USV, is needed.
4. Comparing Shock->Squeak group's freezing data with those from Han et al (2019) data seems questionable since different experimenters ran the experiments and 'blind' observers scored freezing.
5. In Study 2, where rats were freely moving around in the pre-exposure chamber, how was CO2 laser stimulation directed to animals' paws? Was the laser targeting automated or manually adjusted?

Additional comments:

1. Line 634: The statement "While 22 kHz vocalizations have been demonstrated to occur during solitude..." is contradictory of Blanchard et al's (1991, Physiology and Behavior) study which reported NO USV when rats were individually tested with a cat.
2. What is meant by "the sound of freezing" (line 110 and elsewhere)?
3. The 'auto-conditioning' hypothesis needs to be better explained for the general audience.

Reviewer #2 (Remarks to the Author):

Dear Julian Packheiser and co-authors,

Your manuscript "Investigating the mechanistic role of painful self-experience in emotional contagion: an effect of auto-conditioning?" raises very interesting question in the field of the Social Neuroscience of Brains. I've enjoyed reading the manuscript - it is well-written, providing extensively experimental details. But I have several questions before building up final conclusions from the study:

Experiment 1:

Methods:

- When 22kHz vocalization was recorded? It was done during 4 inter-stimulus intervals between exposure of rats to "squeak"/"scrambled squeak"?

Recommendation: for clarity, it is worth to modify Figure 2 by adding timeline for each step of your procedure, e.g. Habituation to Context A session (20 min); "Pre-exposure day 2"/Training session (4 US (shocks) with 4-6 min as ISI. Were animals removed immediately after the last US to handle them for 5 min?

The testing Day 5 consists from 5 "Squeaks"/scrambled Squeaks" (12 min each) - 2-3 min ISI

Results:

- As you recorded freezing 5 times (during playbacks for 12 min) and I assume, 22 kHz calls were recorded 4 times during the ISI, then it will be informative to present dynamics of the freezing response and 22 kHz calls for each "playback" exposure. It could help to determine adaptation in each group, identify the most robust interval with maximal freezing/22kHz calls.

Moreover, authors should perform correlation analysis between Freezing and fearful calls (22 kHz) for each experimental condition. This analysis will be helpful to interpret the obtained data in terms of auto-conditioning, and discuss both indexes of fear response more deeply.

Experiment 2:

Laser-induced pain- it is interesting why laser induced pain response but could not induce fear response assessed by freezing and 22 kHz calls. I'm not sure about "custom" pain scoring system. It is essential if authors will probe pain response on the electrophysiological level using the same set up with laser stimulation, and do recording of neurons of the spinal cord to make sure that experimental parameters are optimal and do induce pain. Again, if compare foot-shock-induced pain and laser-induced pain - how these two set ups biologically comparable?? Even the original paper by Carillo et al., 2019 showed the role of ACC for "empathy" related processes of fear, this information is missing.

That's why it's difficult to conclude about lack of freezing and 22 kHz calls responses in the Experiment 2. Additionally, authors need to modify their experimental design and couple the "Laser group" with "Squeak" as 22kHz calls elicited by rats during the laser-induced pain rather than foot-shock-induced pain.

Reviewer #3 (Remarks to the Author):

The manuscript written by Packheiser et al. investigated the psychological mechanisms of how emotional contagion is formed in rats. They used fear conditioning paradigm and Exp.1 explored how self-emitted distress call (squeak)during training period is associated with experience of electrical footshocks. The exposure of recorded squeak itself did not induce freezing or distress calls, but the preexposure of footshocks sufficiently induced freezing when they heard vocalized calls from conspecifics. The authors argued that it would be attributable to either potential (automated) association learning between shock-induced fear and surrounding sound (distress call) or sensitization to distress calls. Exp.2 aimed to discriminate automated association from sensitization. A repeated exposure of distress calls (recorded and mixed calls) did not induce either freezing or distress calls. However, pain exposure (may be associated with fear) by a CO2 laser was not sufficient to induce distress call (but looks painful) and thus not form an association between distress call and fear (state).

The present experiments seem to be a good challenge to solve a significant problem/topic in psychology. Due to a complexity of factorial analysis, the experimental design and data obtained are needed to be carefully interpreted. There are a couple of major concerns that need to be addressed.

1) Baseline vs. playback

In the day 5, test session, animals were exposed to two phases of stimuli (baseline vs. playback) in Exp.1. If I understand it correctly, the same subjects were exposed to the baseline session and subsequently playback session on day 5 in Exp.1, and on day 7 in Exp.2. In both the cases, exposure of playback calls increased freezing and distress calls regardless of preexposure of footshock or CO2 laser. The authors should mention this effect clearly and discuss this innate (non-conditioned) response. In addition, it may be better to arrange or modify the figures to explicit reaction differences between baseline vs. playback sessions.

2) Laser application vs. footshock exposure

A big problem here is a dissociation between footshock effect and CO2 laser effect on conditioning. It seems like that footshock preexposure is sufficient to form an association with distress calls, but the laser is not. A possible interpretation for it would be that the laser exposure was not sufficient to produce fear response or distress calls in rats and thus no association was formed. It may be that is, although the laser exposure probably induced painful reaction. Another possible interpretation for it is due to animals' innate strategy to form pain-related association. When they received aversive pain stimuli on their paws (e.g., via footshocks), it tends to be easy to form association with floor condition or materials, or surrounding chamber environment. When they received aversive stimuli on their back or fur (e.g., via laser exposure), it may be difficult to form association with surrounding chamber environment. To this point, it is not clear what kinds of factors primarily affect these differences in rats' reactions. The authors need to clarify what kinds of factors they manipulate and affect reactions of rats in test sessions.

3) Dissociation between auto-conditioning vs. sensitization

It should be clarified more carefully a difference between auto-conditioning process and sensitization process. It seems to be that sensitization occurs if reaction is enhanced by repeated exposure of (recorded) distress calls alone. The auto-conditioning may occur when fear state is associated with muscle movement by vocalization (or calls) or external sounds like distress calls. However, if distress call exposure per se induces fear state, it is difficult to distinguish it from auto-conditioning process. Therefore, these hypothesized mechanisms should be defined by operational procedures rather than unclear internal processes. It may be that a point should be discussed here is the existence of self-emitted calls during training.

Reviewer #1 (Remarks to the Author):

Point 1: The authors tested a specific hypothesis that ‘auditory auto-conditioning’ in observer (subject) rats contributes to emotional contagion phenomenon. To test this, study 1 examined three groups: (i) footshock pre-exposed rats tested with auditory playback of squeak (Shock->Squeak), (ii) footshock pre-exposed rats tested with auditory playback of scrambled squeak (Shock->Control), and (iii) no footshock pre-exposed rats tested with auditory playback of squeak (Naïve->Shock). The main findings were that during the auditory playback testing, both Shock->Squeak and Shock->Control groups showed comparably enhanced freezing compared to the Naïve->Shock group but only the Shock->Squeak animals emitted significant 22 kHz USVs. Study 2 then tested fear responses to squeak playback in (i) rats pre-exposed to a CO2 laser-induced pain paired with squeak playback (Laser+Squeak), (ii) CO2 laser-induced pain alone (Laser), (iii) squeak playback alone (Squeak), and (iii) neither CO2 laser nor squeak playback (Naïve) and found no significant group differences. Based on these results, the authors concluded that “We thus demonstrated the sufficiency of pain squeaks to trigger fear in conspecifics in ways that depend on prior shock experience...”

In my view, this is a very specialized study that yielded potentially interesting findings for a narrow audience.

Response: We thank the reviewer for their feedback and want to emphasize that the study’s interest is to a broad audience in the community of social neuroscience. Rats are highly social and a core element of their social behavior repertoire is communication. From the very first days of life, rats communicate primarily through two main communication channels, i.e. the auditory and the olfactory channel. The auditory channel is particularly relevant as it allows for long-distance communication between sender and receiver, e.g. in response to danger, such as predator exposure. Previous studies typically focused on the sender emitting aversive 22 kHz ultrasonic vocalizations reflecting a negative affective state, yet information transfer between sender and receiver was often neglected. This is particularly true for audible squeaks in response to painful stimuli. A better understanding of socio-affective information transfer and processing is needed for multiple reasons. For example, rats are often the first choice in studies aiming at establishing a causal link between brain functions and complex behaviors. It is therefore important to understand in what ways rats from one experimental condition might affect the behavior displayed by rats in other experimental conditions, i.e. in what ways rats are able to exchange relevant information across experimental conditions, obscuring differences between experimental conditions. Obviously, this is not only highly relevant for social behavior studies but also for other studies, such as fear conditioning experiments commonly used to study learning and memory. That said, rats are often the first choice to study complex social behavior phenotypes - not only with relevance to mental disorders characterized by social and communication impairments, such as autism spectrum disorder. In fact, rats appear to be an ideal model system for addressing current limitations of translational research associated with simple standard assays by developing optimized assays for complex social behavior. Our study provides new insights into socio-affective information transfer and processing in rats. Novel findings, such as that audible squeaks leads to 22 kHz ultrasonic calling, can be seen as an important first step in developing optimized assays for complex social behavior that are sensitive enough to reveal impairments in processes that are at the core of mental disorders, such as autism spectrum disorder, e.g. emotional contagion. We therefore believe that our findings have significant implications for very large fields of research and therefore expect that our study is of interest to a broad readership.

Homberg, J. R., Wöhr, M., & Alenina, N. (2017). Comeback of the rat in biomedical research. *ACS Chemical Neuroscience*, 8(5), 900-903.

Homberg, J. R., Adan, R. A., Alenina, N., Asiminas, A., Bader, M., Beckers, T., ... & Genzel, L. (2021). The continued need for animals to advance brain research. *Neuron*, 109(15), 2374-2379.

Point 2: However, the study lacks the following crucial details:

1. Study 1 needs to show freezing, 22-kHz USV and squeak responses in conjunction with four footshocks for two pre-exposed groups. This is necessary to demonstrate the temporal pairing of footshocks with animals' fear responses.

Response: We thank the reviewer for encouraging us to provide further details about the behavior of the animals during pre-exposure in Experiment 1. We now provide the requested data as Figures 3 and 4, and included relevant information in the method and result section of Experiment 1.

In the method we write:

"We also investigated shock-induced changes during the pre-exposure session in more detail by analyzing the rapid temporal dynamics of 22 kHz call emission, squeak production, and freezing behavior from 5 s prior to 10 s after the shock onset. For each of these time periods, if more than 0.5 s of a 1 s bin contained the behavior under investigation, that bin was scored as 1, if not, scored as 0. Proportions for each bin were then calculated separately by pooling all 4 shock trials of all animals that were analyzed for that particular behavior. For 22 kHz calls and squeaks, this analysis was based on a total of ten animals (five later on tested with low amplitude squeaks, and five with low amplitude control stimuli), since we only had ultrasound recordings in these animals during pre-exposure. For freezing, all 24 animals were used. Note that since four shocks are given per animal, the period from -5 to 0 s is not a true baseline, but an interval that in 3/4 of cases occurs after another shock (the inter-shock interval of either 240 or 360 s). Statistical significance of observed proportions were analyzed by comparing each of the observed proportions after shock onset to the average of the proportions observed during the 5 s baseline period via separate binomial tests. The significance level was Bonferroni-corrected at $\alpha=0.05/10$ for each set of analyses."

In the results we write:

"Pre-exposure session

All animals in the Shock->Squeak and Shock->Control groups (19/19) emitted audible squeak vocalizations in response to each of the four footshocks, whereas none of the animals in the NoShock->Squeak group (0/5) emitted any squeaks during their sham pre-exposure session. Freezing differed between the experimental groups (S-W p -value = 0.211; $F_{(2,21)} = 100.01$ $p < 0.001$, $\eta^2 = 0.91$, $BF_{incl} > 100$) and was higher in both the Shock->Squeak and Shock->Control group compared to the NoShock->Squeak group (both p s < 0.001 , $BF_{10s} > 100$, see Figure 3A). There was no difference between the two groups that received shocks ($p = 1.000$, $BF_{10} = 0.52$). This was also the case for 22 kHz recordings in the ten animals for which USV recordings were available ($t_{(8)} = 1.08$, $p = 0.310$, $BF_{10} = 0.70$, see Figure 3B). Descriptive statistics for freezing responses and 22 kHz vocalizations during pre-exposure are presented in Supplementary Table 1 and 2."

Figure 3. Freezing responses and 22 kHz vocalizations during pre-exposure. **A)** Animals that received shocks during pre-exposure (red and yellow) showed higher freezing responses during pre-exposure than animals not receiving shocks (lilac). Freezing was comparable in magnitude for the two shock groups. Freezing was analyzed for all 24 animals tested in Experiment 1. **B)** 22 kHz vocalizations for animals that received shocks during pre-exposure. Note that USVs were recorded for five animals from the Shock->Control and five animals from the Shock->Squeak group only, and no USVs were available for the NoShock group.

“To provide further insights into the timing of the behavioral responses of the animals to the pre-exposure shocks, we analyzed, for each 1 s interval relative to the onset of the shocks, what proportion of trials included squeaks, 22 kHz vocalizations or freezing (Figure 4). We found that all animals emitted squeaks in all trials during the second of shock delivery, but never outside of that epoch. This tight contingency between shocks and squeaks would be ideal for threat-conditioning. In contrast, 22 kHz vocalizations occurred outside of the shock delivery period, and increased significantly 2 s after shock delivery. Freezing was highest before shock delivery, with shock delivery reducing freezing as it triggers more active forms of defensive behavior. This data stems from averaging 4 pre-exposure shocks for each animal, so that the 5 s preceding shocks combine a true baseline period preceding the first shock with the inter-shock interval preceding the other 3 shocks. The freezing rate around 75% prior to shock delivery reflects this averaging of $\frac{1}{4}$ trials with virtually no freezing during the baseline period and $\frac{3}{4}$ of trials with high freezing after the preceding shock.”

Figure 4. Behavioral responses relative to the timing of footshocks during pre-exposure for Experiment 1. A) Exemplar spectrogram in response to the second shock an animal received during the pre-exposure session. The lilac box highlights the squeak occurring during the 1s of shock delivery, while the orange boxes highlight eight 22 kHz vocalizations. Squeaks only occurred during shocks, whereas 22 kHz calls were absent during shocks. **(B)** Proportions of trials during which 22 kHz calls (orange) or squeaks (lilac) were produced as a function of 1 s bins from 5 s before until 10 s after the shock onset. Note that shocks start at $t=0$ and last for 1 s. For each trial, if more than 0.5 s of a particular bin contained a 22 kHz call or a squeak, that bin was scored as 1, if not, scored as 0. Note that the squeaks and 22 kHz calls were only recorded from the subset of 10 animals that had a lower amplitude playback during the playback session, as we had not initially planned to analyze (and hence record) sound emissions during pre-exposure. There were 4 shocks for each of these 10 animals; thus, the proportions were calculated across $10 \times 4 = 40$ total shock trials. Given that 4 shocks are given per animal, the period from -5 to 0 s is not a true baseline, but an interval that in 3/4 of cases occurs after another shock (the inter-shock interval either 240 or 360 s). Error bars indicate standard error of proportions (SEP) as calculated separately for each bin as $SEP = (p(1-p)/n)^{1/2}$, where p is the observed proportion and n is the total number of shock trials. The thick orange bar above the figure indicates significant increases in 22 kHz call emission compared to the baseline period as analyzed by comparing each of the observed proportions after shock onset to the average of the proportions observed during the 5 s baseline period via separate binomial tests (all $ps < 0.005$). Squeaks were observed in all shock trials for all animals during the 1s of footshock delivery, but never outside that bin. **(C)** Same as in (B), but for freezing responses. Note that freezing analysis was conducted across all 24 animals. The thick black bar above the figure indicates significant changes in freezing due to shock exposure.

Unfortunately, for 22 kHz vocalizations, we only recorded USVs during pre-exposure for the ten animals that received a lower amplitude (75 dB) playback later on. Our data however echoes observations we published earlier (Willadsen et al., 2021) that 22 kHz vocalizations become more frequent when footshocks are applied to rats, but that their timing is not tightly linked to the timing of the footshocks (Willadsen et al., 2021). Indeed, 22-kHz USVs start to occur after one, but more typically after two or three shocks (See Figure from Willadsen et al., 2021 copied below). They stop in response to the presentation of external stimuli, such as tone and shock (but reappear soon after their stop). The dynamics of 22 kHz vocalizations as well as freezing in response to shocks has been described in detail in this previous publication (Willadsen et al., 2021).

Willadsen, M., Üngör, M., Sługocka, A., Schwarting, R. K. W., Homberg, J. R., & Wöhr, M. (2021). Fear Extinction and Predictive Trait-Like Inter-Individual Differences in Rats Lacking the Serotonin Transporter. *International Journal of Molecular Sciences*, 22(13), 7088. <https://doi.org/10.3390/ijms22137088>

22-kHz USV and Immobility

B Representative Ethograms | 22-kHz USV and Fear-related Behavior

Figure from Willadsen et al., 2021: Dynamics of freezing/immobility and 22 kHz vocalizations with respect to footshocks. A) Correlation between immobile states and 22 kHz vocalization. B) Ethogram for male and female rats in response to a 20s tone CS (shaded areas) that terminated with a 0.5s 0.5mA footshock. Note how 22 kHz emissions are rare around the first two shocks, and how immobility actually decreases just after the footshocks. Note that the figure separates animals based on sex and serotonin polymorphism in that study.

Point 3: The Pain squeak is defined as those in the audible range (line 91). Fig. 1, however, shows that the squeak spectrogram spans 0-30 kHz and thus includes USV. More information is needed on how squeaks were recorded in relation to footshocks and inter-shock intervals. Fig. 1 only shows 1-sec epoch, which is not too informative.

Response: The reason why we show a 1s period for the spectrogram in Figure 1 is twofold. First, Figure 1 serves to illustrate the properties of the stimulus we played back in the experiment, and not of the behavior of the rats in the experiment. Given that our stimuli were 1s long, it seemed natural to show the spectrogram for the duration of the actual stimulus. Second, we choose to play back 1s stimuli because we have found the squeaks to last 1s when animals receive footshocks of 1s. The latter now become evident from the new Figure 4 reported above, that shows that squeaks - unlike 22 kHz vocalizations - are emitted during the footshock

With regard to referring to the squeaks as in the audible range, the reviewer rightfully points out that pain squeaks, while being audible to the human ear, comprise USV signals outside the human hearing range as well. It is a broadband signal with significant ultrasonic components, but this does not mean that they are ultrasonic vocalizations. Ultrasonic vocalizations are purely in the ultrasonic range. This is not the case for squeaks.

Point 4: Related to comment 2, a crucial control group, i.e., footshock pre-exposed rats tested with auditory playback of 22-kHz USV, is needed.

Response: We thank the reviewer for this suggestion. There are two main reasons why we chose not to include USV playbacks. First, as mentioned above USVs, when emitted, are produced with some delay after shock-experience. This delay, in the order of seconds, would create what is called 'backwards conditioning', where the CS follows the US, and is known to generate a situation in which the CS becomes a signal of safety rather than of threat, and is thus ill suited to explain how rats learn to freeze in response to the distress of others through auto conditioning. We added a section to the introduction to illustrate this point:

"It is worth noting, that the fact that USVs are not emitted during shocks, but a couple of seconds after the shocks, creates an unfavorable situation for auto-conditioning, as protocols in which a tone follows a shock (so called backward conditioning paradigms) are notoriously ineffective, and sometimes even lead to safety learning (e.g., Heth and Rescorla, 1973)."

Heth, C. D., & Rescorla, R. A. (1973). Simultaneous and backward fear conditioning in the rat. *Journal of Comparative and Physiological Psychology*, 82(3), 434.

Furthermore, there is existing evidence that pre-experienced animals freeze to the presentation of USVs (Parsana, Moran et al., 2012, but see Calub et al., 2018), while studies examining the effect of pre-exposure on squeak playback in rats was missing to our knowledge.

Parsana, A. J., Moran, E. E., & Brown, T. H. (2012). Rats learn to freeze to 22-kHz ultrasonic vocalizations through auto-conditioning. *Behavioural Brain Research*, 232(2), 395–399. <https://doi.org/10.1016/j.bbr.2012.03.031>

Calub, C. A., Furtak, S. C., & Brown, T. H. (2018). Revisiting the auto-conditioning hypothesis for acquired reactivity to ultrasonic alarm calls. *Physiology & behavior*, 194, 380–386. <https://doi.org/10.1016/j.physbeh.2018.06.029>

Accordingly, rather than contributing to an existing literature on USVs, The present study focused exclusively on a novel question, namely the role played by pain squeaks - a different type of vocalization that had not yet been appropriately studied. Thus, a control group of 22 kHz vocalizations wouldn't seem necessary in the present paper on squeak playbacks, as this would refer to a different research question that has already been investigated in detail. We therefore did not include this control group and hope that the reviewer agrees with our reasoning.

Point 5: Comparing Shock->Squeak group's freezing data with those from Han et al (2019) data seems questionable since different experimenters ran the experiments and 'blind' observers scored freezing.

Response: We agree with the reviewer that the cross-experimental comparisons should be interpreted with caution due to different experimenters. We added this to the limitation section, which now reads:

"Finally, the cross-experimental comparisons with Han et al. (2019) should be treated with a bit of caution since different experimenters were conducting the experiments which could potentially introduce experimenter effects (Georgiou et al., 2022)."

Georgiou, P., Zanos, P., Mou, T. C. M., An, X., Gerhard, D. M., Dryanovski, D. I., ... & Gould, T. D. (2022). Experimenters' sex modulates mouse behaviors and neural responses to ketamine via corticotropin releasing factor. *Nature Neuroscience*, 25(9), 1191-1200. <https://doi.org/10.1038/s41593-022-01146-x>

Point 6: In Study 2, where rats were freely moving around in the pre-exposure chamber, how was CO2 laser stimulation directed to animals' paws? Was the laser targeting automated or manually adjusted?

Response: The reviewer is correct that rats were allowed to move freely around the pre-exposure chamber. The CO2 laser stimulation was manually directed to the animal's paws with an approximate distance of 10-15 cm between the laser tip and the paw, to allow for consistency between laser deliveries and animals. We added this information to the manuscript.

The new section reads:

"During pre-exposure, the animals could freely explore the arena to prevent any restraining stress. The CO2 laser was applied manually by the experimenter to the hind paw of the animal at an approximate distance of 10 to 15 cm."

Point 7: Additional comments:

1. Line 634: The statement "While 22 kHz vocalizations have been demonstrated to occur during solitude..." is contradictory of Blanchard et al's (1991, *Physiology and Behavior*) study which reported NO USV when rats were individually tested with a cat.

Response: We agree that the study by Blanchard et al. (Physiology and Behavior, 1991) is a very important study that stimulated a lot of research, including research on the audience effect. However, we wish to point out that there are probably hundreds of studies showing that rats emit 22-kHz USV during solitude. Moreover, we wish to point out that the initial findings by Blanchard et al. (1991) on the audience effect were not replicated under more standardized experimental test conditions. This includes the study by Wöhr and Schwarting (Animal Behaviour, 2008), which used a setup and stimuli similar to the ones used in the present study under review.

The finding that 22-kHz calling was not potentiated by a social context in Wöhr and Schwarting may be due to several reasons. One striking difference between the two studies is the aversive stimulus used (cat in Blanchard et al., unlike in the present study, versus foot shocks in Wöhr and Schwarting, like in the present study). Furthermore, the experimental setting differed in multiple ways (open field or visible burrow system versus classical shock chambers). Finally, the sex of the subjects differed (mixed-sex colonies versus male rats only). Importantly, however, the social and nonsocial test conditions of Blanchard et al. (1991) differed not only regarding the absence or presence of conspecifics, but also with respect to the test apparatus, test duration, and housing conditions. It is reasonable to assume that these covarying/confounding factors could have affected behavior. In fact, isolated housing can reduce 22 kHz call production in aversive situations (Inagaki et al. 2004; Nunes Mamede Rosa et al. 2005; Tomazini et al. 2006). The single-housed animals in Blanchard et al.'s (1991) study did not vocalize when exposed to a cat, whereas the socially housed, but individually tested animals in two subsequent studies did vocalize when exposed to a cat (Blanchard et al. 1992; Shepherd et al. 1992). It seems possible, therefore, that the audience effect in Blanchard et al.'s (1991) study is actually based on several different factors, of which two important ones are the absence or presence of cage mates during testing and housing conditions before testing.

We now added to the main text a reference to the 1991 paper: *“While 22 kHz vocalizations have been demonstrated to occur during solitude when exposed to predators (Blanchard et al., 1992, but see Blanchard et al., 1991)”*, but feel that a full discussion of this issue would probably go beyond the scope of this paper.

Blanchard, R. J., Blanchard, D. C., Agullana, R., & Weiss, S. M. (1991). Twenty-two kHz alarm cries to presentation of a predator, by laboratory rats living in visible burrow systems. *Physiology & behavior*, 50(5), 967-972.

Inagaki, H., & Mori, Y. (2014). Relationship between 22-kHz calls and testosterone in male rats. *Hormones and behavior*, 65(1), 42-46.

Rosa, M. L. N. M., Nobre, M. J., Oliveira, A. R., & Brandão, M. L. (2005). Isolation-induced changes in ultrasonic vocalization, fear-potentiated startle and prepulse inhibition in rats. *Neuropsychobiology*, 51(4), 248-255.

Shepherd, J. K., Blanchard, D. C., Weiss, S. M., Rodgers, R. J., & Blanchard, R. J. (1992). Morphine attenuates antipredator ultrasonic vocalizations in mixed-sex rat colonies. *Pharmacology Biochemistry and Behavior*, 41(3), 551-558.

Tomazini, F. M., Reimer, A., Albrechet-Souza, L., & Brandão, M. L. (2006). Opposite effects of short-and long-duration isolation on ultrasonic vocalization, startle and prepulse inhibition in rats. *Journal of neuroscience methods*, 153(1), 114-120.

Wöhr, M., & Schwarting, R. K. (2008). Ultrasonic calling during fear conditioning in the rat: no evidence for an audience effect. *Animal Behaviour*, 76(3), 749-760.

Point 8: What is meant by “the sound of freezing” (line 110 and elsewhere)?

Response: This was an error as the animals conditioned to their own freezing responses rather than the sound of freezing. Since the introduction was re-written to better introduce the auto-conditioning hypothesis, this sentence has been removed.

Point 9: The ‘auto-conditioning’ hypothesis needs to be better explained for the general audience.

Response: We agree with the reviewer and modified the introduction of the manuscript accordingly to provide a detailed description on how auto-conditioning works. It now reads:

“When rats receive noxious footshocks, they emit two types of sounds. One type, called ‘squeaks’, can be heard by humans because they are broadband signals spanning across our audible and ultrasonic range. These squeaks are emitted at very short latency (<50ms), their loudness reflects the intensity of the shock, and continue to be emitted in bouts for as long as the footshocks continue, i.e. for about 1 s when a 1 s footshock is delivered (Jourdan et al., 1995; Carrillo et al., 2019). The other, called 22 kHz ultrasonic vocalization (USV), are beyond human hearing, are narrowband signals with a main frequency typically around 22 kHz, and are not emitted during the footshocks, but typically a few second after the shocks together with freezing (Brudzynski et al., 1993, Choi & Brown, 2003). Unlike the squeaks that are only emitted when rats are in very threatening or painful situations, USVs are emitted under many situations in which rats are in negative affective states, such as predator exposure, aggressive encounters, or long-lasting social isolation - in the laboratory they are typically seen in response to air puff, acoustic startle stimuli, drug withdrawal, or footshocks (Brudzynski et al., 1993). From an information theoretical point of view, that squeaks occur so consistently during footshocks, yet so rarely in other circumstances, makes them a potentially highly informative cue about the pain state of others. From a neural perspective, that rats emit squeaks selectively while they experience footshocks also creates a Hebbian learning opportunity in which nociceptive neurons triggered by the footshocks are repeatedly co-activated with auditory neurons triggered by hearing themselves squeak, and the synapses connecting them could therefore be strengthened to the point where later, hearing a demonstrator squeak would reactivate shock engrams and the associated nocifensive behaviors more strongly -- explaining why pre-exposure could increase the reaction to other individuals being exposed to similar footshocks (Keysers et al., 2014; Keysers & Gazzola, 2014b; Atsak et al., 2011).

From a classical learning perspective, the shock is an unconditioned stimulus (US), hearing themselves squeak a conditioned stimulus (CS), and the US-CS contingency a simultaneous conditioning protocol. Classic studies have demonstrated that roughly simultaneous tone-shock presentations suffice to trigger nocifensive responses upon later presentations of the tone (Heth & Rescorla, 1973) -- although effects are less strong than if the CS precedes and thus predicts the US -- and the word ‘auto-conditioning’ refers to the fact that if the CS is a squeak produced by the animal itself, rather than a tone presented artificially by an experimenter, the animal essentially conditions itself to learn to fear the sounds it has produced in response to a threat (Kim et al., 2010; Parsana, Moran, et al., 2012; Cruz et al., 2020). Together these considerations beg us to test the notion that (a) hearing the squeaks of other animals may suffice to trigger fear responses in a pre-exposed listener and (b) that the contingency between a painful experience and hearing squeaks is what enhances later responses to hearing the squeaks of others, as Hebbian learning and auto-conditioning would suggest. Such auto-conditioning must be contrasted against a

simpler, non-associative explanation based on sensitization alone: footshocks can increase nocifensive reactions to many stimuli independently of a specific contingency between a US and CS (Kamprath & Wotjak, 2004)."

Atsak, P., Orre, M., Bakker, P., Cerliani, L., Roozendaal, B., Gazzola, V., Moita, M., & Keysers, C. (2011). Experience modulates vicarious freezing in rats: A model for empathy. *PLOS ONE*, 6(7), e21855. <https://doi.org/10.1371/journal.pone.0021855>

Brudzynski, S. M., & Ociepa, D. (1992). Ultrasonic vocalization of laboratory rats in response to handling and touch. *Physiology & Behavior*, 52(4), 655–660. [https://doi.org/10.1016/0031-9384\(92\)90393-g](https://doi.org/10.1016/0031-9384(92)90393-g)

Carrillo, M., Han, Y., Migliorati, F., Liu, M., Gazzola, V., & Keysers, C. (2019). Emotional mirror neurons in the rat's anterior cingulate cortex. *Current biology*, 29(8), 1301-1312. e6.

Choi, J.-S., & Brown, T. H. (2003). Central amygdala lesions block ultrasonic vocalization and freezing as conditional but not unconditional responses. *Journal of Neuroscience*, 23(25), 8713–8721.

Cruz, A., Heinemans, M., Marquez, C., & Moita, M. A. (2020). Freezing displayed by others is a learned cue of danger resulting from co-experiencing own freezing and shock. *Current biology*, 30(6), 1128-1135. e6.

Heth, C. D., & Rescorla, R. A. (1973). Simultaneous and backward fear conditioning in the rat. *Journal of Comparative and Physiological Psychology*, 82(3), 434.

Jourdan, D., Ardid, D., Chapuy, E., Eschali r, A., & Le Bars, D. (1995). Audible and ultrasonic vocalization elicited by single electrical nociceptive stimuli to the tail in the rat. *PAIN*, 63(2), 237–249.

Kamprath, K., & Wotjak, C. T. (2004). Nonassociative learning processes determine expression and extinction of conditioned fear in mice. *Learning & memory*, 11(6), 770-786.

Keysers, C., & Gazzola, V. (2014a). Dissociating the ability and propensity for empathy. *Trends in cognitive sciences*, 18(4), 163–166. <https://doi.org/10.1016/j.tics.2013.12.011>

Keysers, C., & Gazzola, V. (2014b). Hebbian learning and predictive mirror neurons for actions, sensations and emotions. *Philosophical Transactions of the Royal Society B: Biological Sciences*, 369(1644), 20130175. <https://doi.org/10.1098/rstb.2013.0175>

Kim, E. J., Kim, E. S., Covey, E., & Kim, J. J. (2010). Social transmission of fear in rats: The role of 22-kHz ultrasonic distress vocalization. *PLOS ONE*, 5(12), e15077.

Parsana, A. J., Moran, E. E., & Brown, T. H. (2012). Rats learn to freeze to 22-kHz ultrasonic vocalizations through auto-conditioning. *Behavioural Brain Research*, 232(2), 395–399. <https://doi.org/10.1016/j.bbr.2012.03.031>

Reviewer #2 (Remarks to the Author):

Dear Julian Packheiser and co-authors,

Your manuscript "Investigating the mechanistic role of painful self-experience in emotional contagion: an effect of auto-conditioning?" raises very interesting question in the field of the Social Neuroscience of Brains. I've enjoyed reading the manuscript - it is well-written, providing extensively experimental details. But I have several questions before building up final conclusions from the study:

Point 1: Experiment 1:

Methods:

- When 22kHz vocalization was recorded? It was done during 4 inter-stimulus intervals between exposure of rats to "squeak"/"scrambled squeak"?

Recommendation: for clarity, it is worth to modify Figure 2 by adding timeline for each step of your procedure, e.g. Habituation to Context A session (20 min); "Pre-exposure day 2"/Training session (4 US (shocks) with 4-6 min as ISI. Were animals removed immediately after the last US to handle them for 5 min?

The testing Day 5 consists from 5 "Squeaks"/scrambled Squeaks" (12 min each) - 2-3 min ISI

Response: We thank the reviewer for suggesting a more complete figure, which we added to Figure 2B in the manuscript (see Response Figure 4). We also added information to both the figure and the text to specify when fear responses were measured. They were recorded during the entire ITI as the reviewer points out. The animals were thus not removed from the chamber after the last stimulus to include an interval for measuring fear responses after the last stimulus. The new section reads:

"Freezing and 22 kHz vocalizations during the auditory playback sessions were quantified during the baseline period (percentage of time spent freezing or vocalizing relative to the total baseline period length, i.e. 12 min) as well as for the playback period (percentage of time spent freezing or vocalizing relative to the playback period length). Since both periods were always 12 minutes long, a freezing value of 100% would thus indicate that animals were freezing for the entire 12 minutes in the respective experimental period. Squeaks during pre-exposure were identified directly from the audio channels of the video recordings, as USV microphones were not present for all animals during this period."

Figure 4. Experimental paradigm and timeline. A) Experiments took place in a two-compartment chamber separated by a transparent divider. Pre-exposure on day 2 could either consist of footshocks (Shock->Squeak and Shock->Control groups) or a resting period (NoShock->Squeak group). Playback on day 5 was of regular squeaks (Shock->Squeak and NoShock->Squeak groups) or phase-scrambled squeaks (Shock->Control). Note that the yellow background in the pre-exposure box B symbolizes that overhead lights were turned on in context B but not A. **B)** Experimental timeline for the auditory playback session on day 5. Baseline freezing and 22 kHz vocalizations were measured across the entire 12 min baseline period. Playback freezing and 22 kHz vocalizations were computed across the entire 12 min playback period. The time-resolved stimulus-by-stimulus presentations of fear responses use the intertrial interval (ITI) between individual squeaks for quantification. Note that ITIs between squeaks were randomly chosen to be either 120s or 180s. The depicted sequence is thus an example of a possible randomization order.

Point 2: Results:

- As you recorded freezing 5 times (during playbacks for 12 min) and I assume, 22 kHz calls were recorded 4 times during the ISI, then it will be informative to present dynamics of the freezing response and 22 kHz calls for each "playback" exposure. It could help to determine adaptation in each group, identify the most robust interval with maximal freezing/22kHz calls. Moreover, authors should perform correlation analysis between Freezing and fearful calls (22 kHz) for each experimental condition. This analysis will be helpful to interpret the obtained data in terms of auto-conditioning, and discuss both indexes of fear response more deeply.

Response: We thank the reviewer for this suggestion and we now performed supplementary analyses. The results however did not reveal patterns that differed from those of the averaged

values reported in the original submission. We therefore decided to move the new figures and results to the supplementary material (Supplementary Figures 1, 5 and 11). The stimulus-by-stimulus resolved figures can be found below.

Supplementary Figure 1. Fear response rates resolved stimulus-by-stimulus over the baseline and playback period for the experimental groups of experiment 1. **A)** Freezing and **B)** 22 kHz vocalizations were quantified over the course of each ITI in between stimulus presentations. Statistics below the graphs reflect the interaction effects between the factors Group and Time (baseline and each ITI between squeak playbacks). Error bars represent the standard error of the mean.

Supplementary Figure 5. Fear response rates resolved stimulus-by-stimulus over the baseline and playback period for the experimental groups of experiment 2. **A)** Freezing and **B)** 22 kHz vocalizations were quantified over the course of each ITI in between stimulus presentations. Statistics below the graphs reflect the interaction effects between the factors Group and Time (baseline and each ITI between squeak playbacks). Error bars represent the standard error of the mean.

Supplementary Figure 11. Fear response rates resolved stimulus-by-stimulus over the baseline and playback period for the experimental groups of experiment 3. **A)** Freezing and **B)** 22 kHz vocalizations were quantified over the course of each ITI in between stimulus presentations. Statistics below the graphs reflect the interaction effects between the factors Group and Time (baseline and each ITI between squeak playbacks). Error bars represent the standard error of the mean.

We also added the requested correlational analyses to the manuscript. For the first and third experiment, we did not find a significant correlation although correlation coefficients were positive. For Experiment 2, we found a significant correlation between freezing and 22 kHz vocalizations during the playback period. Thus, there is some evidence that these variables reflect similar affective states in the animals. As can be seen in Figure taken from Willadsen et al. (2021), animals however also often remain immobile without producing 22 kHz vocalizations which can explain the non-significant correlations. In addition to the correlational analyses, we also used the fear responses during pre-exposure as predictor for fear responses during playback using linear regressions. Indeed, freezing responses were predictive of fear responses during playback in Experiment 1 ($F_{(1,22)} = 6.26, p = 0.020, BF_{10} = 3.08$). No such associations could be found for Experiment 2 and 3, likely because fear responses during pre-exposure were marginal.

The new methods section for Experiment 1 reads:

“Since we measured freezing for all animals as well as 22 kHz vocalizations for 10 animals during pre-exposure, we used this as a predictor for the freezing response or 22 kHz vocalizations during playback to identify whether a stronger aversive experience during shock pre-exposure also leads to higher fear response during playback. In addition, we correlated freezing responses and 22 kHz vocalizations during pre-exposure and playback to identify if these measures are associated with each other. As for the ANOVAs, we used both frequentist and Bayesian regression/correlation to provide a comprehensive overview.”

The new results section for Experiment 1 reads:

“Linear regression analyses revealed that freezing during pre-exposure was predictive of freezing during the playback ($F_{(1,22)} = 6.26, p = 0.020, BF_{10} = 3.08$). This was not the case for 22 kHz vocalizations

however ($F_{(1,8)} = 1.32$, $p = 0.283$, $BF_{10} = 0.73$). There was no significant correlation between the emission of 22 kHz vocalizations and freezing responses during squeak playback ($r_{(23)} = 0.33$, $p = 0.117$, $BF_{10} = 0.89$, Spearman correlation).

The new methods section for Experiment 2 reads:

“As in Experiment 1, we also used freezing responses during pre-exposure as a predictor for freezing during playback. Since we also recorded 22 kHz vocalizations during pre-exposure in this experiment, the same procedure was applied for this measure of discomfort. Finally, we correlated freezing responses and 22 kHz vocalizations during playback to identify if these measures are associated with another.”

The new section for Experiment 2 reads:

“Using linear regression, we found that freezing during pre-exposure was not predictive of freezing during the playback ($F_{(1,73)} = 1.21$, $p = 0.275$, $BF_{10} = 0.40$). The same was true for 22 kHz vocalizations across animals ($F_{(1,73)} = 1.04$, $p = 0.311$, $BF_{10} = 0.37$). There was a significant correlation between the emission of 22 kHz vocalizations and freezing responses during squeak playback ($r_{(76)} = 0.46$, $p < 0.001$, $BF_{10} > 100$, Spearman correlation).”

The new methods section for Experiment 3 reads:

“We also used the fear responses during pre-exposure as predictor for the fear responses during playback to identify if rats with stronger fear responses during pre-exposure would react more strongly during pain squeak playback. Finally, we correlated freezing responses and 22 kHz vocalizations during pre-exposure and playback to identify if these measures are associated with another.”

The section for Experiment 3 reads:

“To test whether animals for whom the pre-exposure might have been more distressing would later respond more intensely to playback, we examined whether there was an association between individual differences in behavior during pre-exposure and playback. Using linear regression, we found that freezing during pre-exposure was not predictive of freezing ($F_{(1,18)} = 0.26$, $p = 0.619$, $BF_{10} = 0.44$) during playback. The results were similar for 22 kHz vocalizations ($F_{(1,18)} = 0.58$, $p = 0.455$, $BF_{10} = 0.49$). Since we could measure squeaking during pre-exposure in the Asynchronous group and observed occasional squeaks, we correlated the number of emitted squeaks during pre-exposure with the fear responses during playback to identify if more squeaking during pre-exposure was associated with higher freezing responses or 22 kHz vocalizations upon playback. We did not find an association for either variable in the Asynchronous group (freezing: $r_{(9)} = -0.47$, $p = 0.173$, $BF_{10} = 0.94$; 22 kHz USVs: $r_{(9)} = 0.24$, $p = 0.501$, $BF_{10} = 0.56$, Spearman correlations). The correlation between the emission of 22 kHz vocalizations and freezing responses during squeak playback did not reach significance but showed anecdotal evidence in favor of the alternative hypothesis ($r_{(19)} = 0.37$, $p = 0.105$, $BF_{10} = 1.42$, Spearman correlation).”

Point 3: Experiment 2:

Laser-induced pain- it is interesting why laser induced pain response but could not induce fear response assessed by freezing and 22 kHz calls. I'm not sure about "custom" pain scoring system. It is essential if authors will probe pain response on the electrophysiological level using the same set up with laser stimulation, and do recording of neurons of the spinal cord to make sure that experimental parameters are optimal and **do induce pain**. Again, if compare **foot-shock-induced**

pain and laser-induced pain - how these two set ups biologically comparable?? Even the original paper by Carillo et al., 2019 showed the role of ACC for "empathy" related processes of fear, this information is missing.

That's why it's difficult to conclude about lack of freezing and 22 kHz calls responses in the Experiment 2. Additionally, authors need to modify their experimental design and couple the "Laser group" with "Squeak" as 22kHz calls elicited by rats during the laser-induced pain rather than foot-shock-induced pain.

Response: We thank the reviewer for raising this central issue: do we have evidence that the application of the CO2 laser produced pain? Clearly, if it didn't produce pain, our experiment is not a test of auto-conditioning, which would require pairing an experience of pain with hearing a squeak. It is thus essential to use the most established indicator available in rodents to determine if the intensity of CO2 laser we used here was above the pain threshold or not.

A recent review of the available literature on methods to measure pain in rodents (Deuis, Dvorakova, & Vetter 2017) summarizes the state of the field as follows: "Pain cannot be directly measured in animals; instead pain is inferred from "pain-like" behaviors, such as withdrawal from a nociceptive stimulus, which is the most commonly used method to quantify nociception in animal studies." It further states that paw withdrawal following thermal stimuli applied to the hind paw are a particularly good index of pain, as paw withdrawal following thermal stimulation is strongly reduced following spinal cord transection (Giglio et al., 2006), and therefore does measure higher level processing in the brain that is more likely to be associated with the affective valence of pain. Specifically, they advise to determine the pain-threshold by determining an intensity of stimulation that triggers paw withdrawal in at least 50% of trials. In our paper, we therefore use this most widely accepted method to determine if the CO2 laser application we used surpassed the pain threshold. Many apply the methods in a binary way (does or does not trigger paw withdrawal). We attempted to provide a more nuanced scale, that differentiates a score=2 when the paw is withdrawn, and a score=3 when the entire rat moves away. Scores below 2 are not considered sufficient to consider that the stimulus surpassed the pain threshold. Unfortunately, the way we presented our pain score, in its cumulative form, does not allow the reader to appreciate directly, whether at least 50% of trials lead to paw- or full-body withdrawal. We have therefore amended the manuscript to also present the proportion of trials leading to a paw- or full-body withdrawal, which now reads:

"In a first step, we compared responses to our customized pain reaction scale between all experimental groups that either received a high (Laser and Laser+Squeak) or low intensity laser stimulation (Squeak and Naive; Figure 7A left). Because the most frequently used criterion to determine whether a stimulus is above the pain threshold is to determine whether a given trial did or did not lead to paw withdrawal, and consider stimuli that trigger such withdrawal in at least 50% of trials to be above pain threshold (Deuis et al., 2017) we also analyzed our data in terms of the number of trials (out of the possible 4), in which each animals withdrew their paw (Figure 7A right). A one-way ANOVA revealed a highly significant difference across groups ($F_{(3,68)} = 211.02, p < 0.001, BF_{10} > 100$), with post hoc tests revealing that this was due to the two conditions with Laser triggering a similar number of withdrawals ($t = 1.91, p = 0.23, BF_{10} = 1.09$) that was significantly higher than that for the two conditions without laser (all $t_s > 16.80$, all $p_s < 0.001$, all $BF_{10s} > 100$), which in turn were similar ($t = 0.30, p = 0.990, BF_{10} = 0.34$). Importantly, the median number of pain-like behavioral responses (as defined in the literature as paw-withdrawal) was zero for the two conditions without laser, and 4 for the conditions with laser. If one uses a 50% response threshold to determine whether a stimulus intensity was above or below the pain threshold, this provides strong evidence that the application of the laser did trigger pain in the Laser or Laser+Squeak groups but not in the Naive or Squeak only groups. Whether the pain level was similar

to that triggered by footshocks is doubtful, as pain squeaks, considered evidence for relatively intense pain (Mogil, 2009), was not observed in any of the animals during pre-exposure in Experiment 2.”

Deuis, J. R., Dvorakova, L. S., & Vetter, I. (2017). Methods used to evaluate pain behaviors in rodents. *Frontiers in molecular neuroscience*, 10, 284.

Mogil, J. S. (2009). Animal models of pain: progress and challenges. *Nature Reviews Neuroscience*, 10(4), 283-294.

Figure 7. Pain and fear responses during pre-exposure on day 2. **A)** Cumulative pain responses to the laser for the four experimental groups (left), as well as the number of trials (amongst the four for each animal) that triggered paw-withdrawal or a full body escape. Descriptive values for pain scores can be found in Supplementary Table 5. **B)** Proportion of freezing and C) proportion of 22 kHz calls for the different experimental groups.

The reviewer invites us to record from the spinal cord to establish that the CO2 laser triggered pain. However, we feel that spinal cord recordings are not the most compelling evidence for the CO2 to trigger pain for two reasons: (a) Spinal cord recordings are a very unusual criterion to determine whether a stimulus exceeded the pain threshold, while paw retraction is the most frequently used and established criterion; and (b) pain, as opposed to mere nociception, is thought to depend on an affective component of pain that is associated with supra-spinal processes, such as cingulate cortex processing, making spinal responses an atypical index of the suffering associated with pain. We have already demonstrated that similar CO2 laser intensities delivered to the same body parts do trigger cingulate responses (Carrillo et al., 2019) that have been associated with the affective component of pain, although such reverse inference is always difficult, and would also be difficult for spinal cord recordings.

Carrillo, M., Han, Y., Migliorati, F., Liu, M., Gazzola, V., & Keyser, C. (2019). Emotional mirror neurons in the rat’s anterior cingulate cortex. *Current biology*, 29(8), 1301-1312. e6.

The reviewer rightly points out that laser-induced pain and pain via shock delivery are perhaps qualitatively different experiences that call the comparability of experiment 1 and 2 into question. We therefore conducted a follow-up experiment that mimics the principle of the laser experiment (painful experience without intrinsic squeak production) but using footshocks delivered at an

intensity tailored not to trigger squeaks. Using pilot data, we determined that not all footshocks invariably result in squeaks as animals receiving 0.2mA footshocks will only rarely squeak, even at longer shock durations. We added a section to the discussion of Experiment 2 to motivate the new experiment. It reads:

“A clear limitation of the laser-induced pain is that a direct comparison between Experiment 1 and 2 is questionable despite the otherwise similar experimental protocols since the modality of the painful experience is very different between laser and shock exposure. Since we could also not find any differences between the experimental groups receiving laser pain and the control groups, we conducted a further experiment to disentangle the contribution of auto-conditioning and sensitization and increase the comparability to Experiment 1. Although the exposure to shocks generally induces squeaks, the invariance of squeaking can be modulated through the intensity of the electrical stimulus. Thus, a new experiment was conceived using low intensity shocks to induce an aversive experience without eliciting pain squeaks in the animals. To dissociate between auto-conditioning and sensitization effects, we used comparable procedures as in Experiment 2 by using squeak playback during shock pre-exposure.”

A new experiment was designed using these sub-squeak, milder footshocks, to explore whether a stimulus that is perhaps qualitatively more similar to those of Experiment 1, but mild enough not to trigger squeaks, could be used to condition an animal to squeak-playbacks. One group heard squeaks during shock delivery (Synchronous group). The other group heard a squeak playback during the shock ITI (Asynchronous group). We predicted that, if auto-conditioning can occur in response to these milder footshocks, fear responses should be increased during subsequent squeak playback in the synchronous group. If fear responses do not differ between the synchronous and asynchronous group, it would rather speak in favor of generalized sensitization. We kept all conditions identical to experiment 1 to allow for maximum comparability (i.e. number of mild shocks during pre-exposure, number of squeak playbacks, duration of experimental phases). The paradigm is shown below.

Figure 9. Experimental paradigm of Experiment 3. Experimental procedures on day 1, 3, 4 and 5 were kept identical to Experiment 1. On day 2 however, we presented all animals with low amplitude shocks (0.2mA, 4s. 0.2mA was chosen in a pilot study as a threshold intensity that triggered squeaks in only rare instances) that were either synchronously paired with 4s of pain squeak playback that covered the entire period of shock delivery or asynchronously, with the 4s squeak playback presented exactly in the middle of the 240-360s interval between two shocks.

During pre-exposure, we observed 11 squeaks from 40 trials (27.5%) across 10 animals in the asynchronous group, demonstrating that shock intensity was at the upper limit to still provide a meaningful difference between groups. To assess whether mild shocks induced fear responses,

we compared this group to the squeak only group from experiment 2 as a control pre-exposure group that also had squeak playback but no aversive events. Freezing and 22 kHz vocalizations during pre-exposure did not differ between the synchronous and asynchronous group.

The pre-exposure result section reads:

“Pre-exposure (Day 2)

We again first compared fear responses between the Synchronous and Asynchronous group during the mild shock pre-exposure. We found anecdotal to moderate evidence against a difference in freezing (S-W p-value < 0.001; $W = 38$, $p = 0.393$, $BF_{10} = 0.51$, Figure 10A) as well as 22 kHz calls (S-W p-value = 0.008; $W = 41$, $p = 0.428$, $BF_{10} = 0.43$, Figure 10B) during pre-exposure. In total, we presented 40 shocks to animals in the Asynchronous group ($n = 10$ animals x 4 shocks). The animals in the Asynchronous group squeaked in 11/40=27.5% of shocks suggesting that the shock intensity could not have been any higher without compromising the experimental manipulation. Squeaking in the Synchronous group could not be measured as they occurred simultaneous to the squeak playback. Descriptive statistics for freezing responses and 22 kHz call emissions during pre-exposure are presented in Supplementary Tables 12 and 13, respectively.”

During playback, we observed significantly increased freezing and 22 kHz vocalizations compared to baseline suggesting that the animals showed some fear responses in response to the stimuli. When comparing both groups during the playback period however, no significant differences were observed. In fact, the data rather suggested evidence for the null hypothesis supporting the notion that observed effects were due to sensitization. It needs to be noted, however, that such mild shocks in general induced very low fear responses during pre-exposure. We also could not find a quantitative relation between how much animals vocalized or froze during pre-exposure and fear responses during playback. This calls into question whether stimuli of an intensity low enough not to cause systematic squeaks allow us at all to study the mechanisms at play during the exposure to shock intensities typically used for threat conditioning. Since squeaking was already observed at 0.2mA occasionally, increasing intensity further would have precluded to disentangle shock induced nociception and squeak re-afference, creating a conundrum, in which 0.2mA shocks are unsuited because they are too mild to produce fear conditioning, but more intense shocks are also unsuited, because they trigger too many squeaks to dissociate the threat from the squeak-reafference.

Figure 10. Fear responses during pre-exposure. Neither A) freezing nor B) 22 kHz vocalizations differed between the experimental groups during pre-exposure to low amplitude shocks.

The playback result section reads:

“Squeak Playback (Day 5)

During the squeak playback session on day 5, we again first investigated baseline differences between the experimental groups in fear responses. We found absence of evidence for freezing (S-W p-value < 0.001; $W = 74, p = 0.052, BF_{10} = 0.87$) and anecdotal to moderate evidence against a difference between groups for 22 kHz vocalizations (S-W p-value < 0.001; $W = 50, p = 1.000, BF_{10} = 0.40$). During the playback, we found anecdotal to moderate evidence against group differences for freezing (S-W p-value = 0.273; $t_{(18)} = 1.32, p = 0.203, BF_{10} = 0.51$) and absence of evidence for 22 kHz vocalizations (S-W p-value < 0.001; $W = 31.5, p = 0.093, BF_{10} = 0.72$). The same results were found for the difference scores between playback and baseline (freezing: S-W p-value = 0.592; $t_{(18)} = 0.83, p = 0.417, BF_{10} = 0.73$; 22 kHz calls: S-W p-value < 0.001; $W = 29, p = 0.069, BF_{10} = 0.88$).

To test whether animals for whom the pre-exposure might have been more distressing would later respond more intensely to playback, we examined whether there was an association between individual differences in behavior during pre-exposure and playback. Using linear regression, we found that freezing during pre-exposure was not predictive of freezing ($F_{(1,18)} = 0.26, p = 0.619, BF_{10} = 0.44$) during playback. The results were similar for 22 kHz vocalizations ($F_{(1,18)} = 0.58, p = 0.455, BF_{10} = 0.49$). Since we could measure squeaking during pre-exposure in the Asynchronous group and observed occasional squeaks, we correlated the number of emitted squeaks during pre-exposure with the fear responses during playback to identify if more squeaking during pre-exposure was associated with higher freezing responses or 22 kHz vocalizations upon playback. We did not find an association for either variable in the Asynchronous group (freezing: $r_{(9)} = -0.47, p = 0.173, BF_{10} = 0.94$; 22 kHz USVs: $r_{(9)} = 0.24, p = 0.501, BF_{10} = 0.56$, Spearman correlations). The correlation between the emission of 22 kHz vocalizations and freezing responses during squeak playback did not reach significance but showed anecdotal evidence in favor of the alternative hypothesis ($r_{(19)} = 0.37, p = 0.105, BF_{10} = 1.42$, Spearman correlation).

Results for each individual experimental group are depicted in Figure 11A for freezing and Figure 11B for 22 kHz calls. Descriptive statistics for freezing rates and 22 kHz calls during squeak playback are presented in Supplementary Tables 14 and 15, respectively. A time-resolved figure representing freezing responses and 22 kHz calls for each stimulus presentation can be found in Supplementary Figure 11.”

Figure 11. Fear responses during playback. A) We found no differences in freezing during baseline or playback between the two experimental groups. Direct comparisons between baseline and playback phase can be found in Supplementary Figure 12. B) The same result pattern was found for 22 kHz vocalizations. Direct comparisons between baseline and playback phase can be found in Supplementary Figure 13.

To accommodate the new results, we modified the discussion section and integrated the findings from the third experiment. The new sections read:

“Due to the inconclusiveness of the results from Experiment 2 and the modality change from shock to laser, we conducted a third experiment that procedurally mimicked the experimental design from Experiment 2 but used low amplitude footshocks instead of a CO₂ laser to trigger an

unpleasant sensation qualitatively similar to the footshocks of Experiment 1 but low enough in intensity not to systematically trigger squeaks. As in Experiment 2, we then played back squeaks during pre-exposure. Specifically, we synchronized the squeak playback to the shock in our experimental group, to mimic auto-conditioning. In the control group, we still played back the squeaks, but asynchronously, so as to have a similar opportunity for sensitization while perturbing the potential for the associative learning central to auto-conditioning. Similar to Experiment 2, there was no observable difference between the two groups. However, fear responses during pre-exposure were also rather mild, calling into question if low amplitude foot shocks triggered an aversive state in the animals sufficient to test the contribution of auto-conditioning during the experience of the more threatening shocks typically used for pre-exposure in the literature and in Experiment 1.”

“Although the second experiment and the third experiment had the specific aim to shed light onto the nature of the process that accounts for increased vicarious freezing in the literature (Atsak et al., 2011; Han et al., 2019; Terranova et al., 2022) and that generated the results from our first experiment, the results remain unfortunately inconclusive. While the results of both experiments speak at first glance against the auto-conditioning and in favor of the sensitization hypothesis, the absence of pronounced fear responses in both experiments when presented with the laser or a low amplitude shock call into question if any fear responses observed during playback were due to sensitization to aversive stimuli at all.”

Atsak, P., Orre, M., Bakker, P., Cerliani, L., Roozendaal, B., Gazzola, V., Moita, M., & Keysers, C. (2011). Experience modulates vicarious freezing in rats: A model for empathy. *PLOS ONE*, 6(7), e21855. <https://doi.org/10.1371/journal.pone.0021855>

Han, Y., Bruls, R., Soyman, E., Thomas, R. M., Pentaraki, V., Jelinek, N., Heinemans, M., Bassez, I., Verschooren, S., & Pruis, I. (2019). Bidirectional cingulate-dependent danger information transfer across rats. *PLoS biology*, 17(12), e3000524.

Terranova, J. I., Yokose, J., Osanai, H., Marks, W. D., Yamamoto, J., Ogawa, S. K., & Kitamura, T. (2022). Hippocampal-amygdala memory circuits govern experience-dependent observational fear. *Neuron*, 110(8), 1416-1431.e13. <https://doi.org/10.1016/j.neuron.2022.01.019>

“Taking the results from Experiment 2 and 3 together, it seems unlikely that the null effects from the second experiment were due to changes in modality (CO2 vs footshocks). Rather, it seems that the painful experience was too low and this apparently prevented forming an association with fear-related cues such as squeaks. Such results are in line with dose-dependent studies on fear conditioning in rats that showed no conditioned fear response for a 0.2 mA shock but reliable conditioned fear responses from 0.5 mA onwards (Wöhr et al., 2005). We tried to increase the potential for fear conditioning of 0.2 mA shocks by increasing the duration to 4s as the study of Wöhr et al. (2005) used 500ms duration shocks. This however did not seem to increase fear responses during playback. It thus seems likely that testing the auto-conditioning hypothesis for squeaks on a strictly behavioral level faces a conundrum that may very well make the approach impossible: from an experimental point of view, the nociceptive stimulus has to be mild enough not to trigger squeaks systematically, in order to then compare animals with an added squeak playback against those without squeaks, yet, from a threat-conditioning point of view, the noxious stimulus has to be threatening enough, that it actually will produce a squeak. Thus, future studies could complement our findings using more invasive techniques by for example temporally deafening the animals during shock pre-exposure using pharmacological injections.”

Wöhr, M., Borta, A., & Schwarting, R. K. (2005). Overt behavior and ultrasonic vocalization in a fear conditioning paradigm: a dose–response study in the rat. *Neurobiology of learning and memory*, 84(3), 228-240.

Reviewer #3 (Remarks to the Author):

The manuscript written by Packheiser et al. investigated the psychological mechanisms of how emotional contagion is formed in rats. They used fear conditioning paradigm and Exp.1 explored how self-emitted distress call (squeak)during training period is associated with experience of electrical footshocks. The exposure of recorded squeak itself did not induce freezing or distress calls, but the preexposure of footshocks sufficiently induced freezing when they heard vocalized calls from conspecifics. The authors argued that it would be attributable to either potential (automated) association learning between shock-induced fear and surrounding sound (distress call) or sensitization to distress calls. Exp.2 aimed to discriminate automated association from sensitization. A repeated exposure of distress calls (recorded and mixed calls) did not induce either freezing or distress calls. However, pain exposure (may be associated with fear) by a CO2 laser was not sufficient to induce distress call (but looks painful) and thus not form an association between distress call and fear (state).

The present experiments seem to be a good challenge to solve a significant problem/topic in psychology. Due to a complexity of factorial analysis, the experimental design and data obtained are needed to be carefully interpreted. There are a couple of major concerns that need to be addressed.

Point 1: Baseline vs. playback

In the day 5, test session, animals were exposed to two phases of stimuli (baseline vs. playback) in Exp.1. If I understand it correctly, the same subjects were exposed to the baseline session and subsequently playback session on day 5 in Exp.1, and on day 7 in Exp.2. In both the cases, exposure of playback calls increased freezing and distress calls regardless of preexposure of footshock or CO2 laser. The authors should mention this effect clearly and discuss this innate (non-conditioned) response. In addition, it may be better to arrange or modify the figures to explicit reaction differences between baseline vs. play back sessions.

Response: We fully agree with the reviewer that the innate response to our auditory stimuli should be discussed further. In experiment 2, there were baseline to playback differences even in groups without pre-exposure to harmful and aversive stimuli. The reviewer suggests modifying the figures to focus on differences between baseline and playback rather than group differences. Our initial manuscript already included these figures in the supplementary material. This figure also included additional statistics that quantified differences between baseline and playback period directly. Since our main focus of the manuscript was on differences between the experimental groups, we prefer to leave these figures in the supplement. We however discussed the baseline fear responses to squeaks and scrambled squeaks in more detail in the discussion. It reads:

“In contrast to the first experiment, there was substantial freezing in experiment 2 after the presentation of squeaks as well as scrambled squeaks suggesting that loud noises are sufficient to trigger freezing in Long Evans rats despite them not having been exposed to an aversive and harmful stimulus before. There were however barely any 22 kHz vocalizations in both experiments during playback suggesting that these freezing responses may reflect an affective state that differs from the negative affective state triggered in shock pre-exposed animals in Experiment 1.”

An exemplary figure from the supplementary material is attached in the response letter.

Supplementary Figure 2. Proportion of freezing responses in percent during baseline and auditory playback in the playback session. Paired tests (t-test, if the Shapiro-Wilk (S-W) normality test did not reject normality, i.e. S-W p -value > 0.05 or Wilcoxon signed-rank, if it did, i.e. S-W $p < 0.05$) indicated that both the **A**) Shock \rightarrow Squeak group (S-W p -value = 0.773; $t_{(9)} = 4.26$, $p = 0.002$, $BF_{10} = 21.90$) and the **B**) Shock \rightarrow Control group (S-W p -value = 0.666; $t_{(8)} = 3.86$, $p = 0.005$, $BF_{10} = 11.41$) increased their freezing from baseline to the playback period whereas the **C**) NoShock \rightarrow Squeak group showed a trend towards a significant increase (S-W p -value = 0.207; $t_{(4)} = 2.52$, $p = 0.066$, $BF_{10} = 1.83$). Animals with higher amplitude playback are marked by an open circle. The p -values indicated for the changes from baseline to playback were not corrected for multiple comparisons, but applying an FDR correction would not alter the conclusion.

Point 2: Laser application vs. footshock exposure

A big problem here is a dissociation between footshock effect and CO2 laser effect on conditioning. It seems like that footshock preexposure is sufficient to form an association with distress calls, but the laser is not. A possible interpretation for it would be that the laser exposure was not sufficient to produce fear response or distress calls in rats and thus no association was formed. It may be that is, although the laser exposure probably induced painful reaction.

Another possible interpretation for it is due to animals' innate strategy to form pain-related association. When they received aversive pain stimuli on their paws (e.g., via footshocks), it tends to be easy to form association with floor condition or materials, or surrounding chamber environment. When they received aversive stimuli on their back or fur (e.g., via laser exposure), it may be difficult to form association with surrounding chamber environment.

Response: We fully agree with the reviewer that the formation of associations might have been different between experiment 1 and 2. To elucidate whether it was a question of modality or aversiveness of the pre-exposure procedure, we added a new experiment that used low amplitude shocks that rarely result in squeaks (0.2mA). The experiment used two different experimental groups that either received synchronous squeak playback during these low amplitude shocks or squeak playback during the ITI.

Figure 9. Experimental paradigm of Experiment 3. Experimental procedures on day 1, 3, 4 and 5 were kept identical to Experiment 1. On day 2 however, we presented all animals with low amplitude shocks (0.2mA, 4s. 0.2mA was chosen in a pilot study as a threshold intensity that triggered squeaks in only rare instances) that were either synchronously paired with 4 s of pain squeak playback that covered the entire period of shock delivery or asynchronously, with the 4 s squeak playback presented exactly in the middle of the 240-360 s interval between two shocks. .

We found comparable results to experiment 2 with negligible fear responses during pre-exposure and during playback of squeaks (see response to Reviewer 2 for details). In the light of this new experiment, we feel that the problem was not a qualitative difference between shocks in Experiment 1 and laser in Experiment 2, with only the shocks forming a pain-related association but rather a quantitative difference: the CO2 laser or the 0.2mA shocks did not reach the necessary level of threat or pain for a robust association to be formed. We thus believe that a minimum level of pain/threat is necessary that was neither achieved using the focal application of the laser nor via low amplitude footshocks. We added the following section to the discussion as it might be impossible to test the auto-conditioning hypothesis with respect to pain squeaks on the behavioral level if the level of pain must reach a certain threshold to form an association and that this threshold is invariantly linked to squeaking. It reads:

“Taking the results from Experiment 2 and 3 together, it seems unlikely that the null effects from the second experiment were due to changes in modality (CO2 vs footshocks). Rather, it seems that the painful experience was too low and this apparently prevented forming an association with fear-related cues such as squeaks. Such results are in line with dose-dependent studies on fear conditioning in rats that showed no conditioned fear response for a 0.2 mA shock but reliable conditioned fear responses from 0.5 mA onwards (Wöhr et al., 2005). We tried to increase the potential for fear conditioning of 0.2 mA shocks by increasing the duration to 4s as the study of Wöhr et al. (2005) used 500ms duration shocks. This however did not seem to increase fear responses during playback. It thus seems likely that testing the auto-conditioning hypothesis for squeaks on a strictly behavioral level faces a conundrum that may very well make the approach impossible: from an experimental point of view, the nociceptive stimulus has to be mild enough not to trigger squeaks systematically, in order to then compare animals with an added squeak playback against those without squeaks, yet, from a threat-conditioning point of view, the noxious stimulus has to be threatening enough, that it actually will produce a squeak. Thus, future studies could complement our findings using more invasive techniques by for example temporally deafening the animals during shock pre-exposure using pharmacological injections.”

Wöhr, M., Borta, A., & Schwarting, R. K. (2005). Overt behavior and ultrasonic vocalization in a fear conditioning paradigm: a dose–response study in the rat. *Neurobiology of learning and memory*, 84(3), 228-240.

To this point, it is not clear what kinds of factors primarily affect these differences in rats' reactions. **The authors need to clarify what kinds of factors they manipulate and affect reactions of rats in test sessions.**

We hope that our inclusion of Experiment 3 now constrains this question, suggesting that the problem is that noxious stimuli weak enough not to trigger squeaks systematically are also unfortunately too weak to serve as US in threat conditioning.

Point 3: Dissociation between auto-conditioning vs. sensitization

It should be clarified more carefully a difference between auto-conditioning process and sensitization process. It seems to be that sensitization occurs if reaction is enhanced by repeated exposure of (recorded) distress calls alone. The auto-conditioning may occur when fear state is associated with muscle movement by vocalization (or calls) or external sounds like distress calls. However, if distress call exposure per se induces fear state, it is difficult to distinguish it from auto-conditioning process. Therefore, these hypothesized mechanisms should be defined by operational procedures rather than unclear internal processes. It may be that a point should be discussed here is the existence of self-emitted calls during training.

Response: We thank the reviewer for highlighting this issue that relates to the previous point on innate fear responses upon hearing squeaks or squeak-like events. We fully agree that a dissociation between the auto-conditioning and sensitization hypothesis is difficult since both processes can be at play simultaneously. We want to stress that our second and newly added third experiment were however specifically designed to disentangle sensitization (increased fear responses to novel stimuli following exposure to a harmful and traumatic event) and auto-conditioning (one's own fear/pain responses during a harmful or traumatic event become a conditioned stimulus). Thus, these concepts are not abstract notions but testable hypotheses in both these experiments. Since we found no specific increase in fear responses after squeak playback in the Laser+Squeak group (Exp 2) and Synchronous group (Exp 3), it seems intuitive to conclude that our results are due to sensitization. However, the observed fear responses in both experiments for all groups with a painful pre-exposure (laser in Exp 2 and mild shock in Exp3) did not exceed innate fear responses to loud noises in general. Thus, we cannot ultimately conclude which hypothesis is correct from these behavioral experiments alone in line with our response to the reviewer's second comment. To clarify on the auto-conditioning and sensitization hypothesis, we added more information to the discussion of experiment 1. The new sections read:

"According to the auto-conditioning hypothesis, the self-emitted squeaks become a CS that associates with the shocks as a US during pre-exposure. Hearing these squeaks in conspecifics then later triggers the conditioned response."

"Playing back the scrambled squeaks elicited a similar level of freezing as regular squeak playback but lower levels of 22 kHz calls. Such responses indicate that not only auto-conditioning (that should be more specific to the self-emitted intact squeaks) but also sensitization to auditory stimuli more generally (including the phase-scrambled squeaks) might have been at play after being exposed to aversive stimuli. Sensitization to fear occurs after a harmful and possibly traumatic event that causes subsequently elevated fear and stress responses under conditions that would normally not trigger such responses (Poulos et al., 2015)."

Poulos, A. M., Zhuravka, I., Long, V., Gannam, C., & Fanselow, M. (2015). Sensitization of fear learning to mild unconditional stimuli in male and female rats. *Behavioral neuroscience*, 129(1), 62.

Reviewers' comments:

Reviewer #1 (Remarks to the Author):

I appreciate the detailed responses and data analyses to address my comments and concerns. However, I remain perplexed by the authors' interpretation of their findings that pre-exposure to footshocks leads to auto-conditioning to self-generated squeaks that can underlie emotional contagion.

During the shock pre-exposure procedure (which involves four unpredictable 1 sec, 0.8 mA footshocks with an intershock interval of 240-360 sec), each shock US elicits a tightly coupled squeak response. This squeak response spans both audible and ultrasound ranges within the same 1 sec period, as depicted in Fig. 4A, which is essentially a UR. Subsequent to the footshock, 22-kHz USV calls arise within the intershock interval.

Unless I have misunderstood, the authors argue that each squeak acts as a CS undergoing autoconditioning to the shock US. Yet, considering the shock US and its evoked squeak response transpire almost simultaneously (with the sequence being shock US followed closely by squeak CS), how can fear conditioning occur under such simultaneous (backward) conditioning conditions?

On the other hand, the USV emitted during the intershock interval seems more plausible as a CS autoconditioned to the subsequent shock US. For instance, the USV #1, following footshock #1, might serve as a CS preceding footshock #2 (indicative of forward conditioning rather than backward conditioning). Similarly, USV #2 could precede footshock #3, and USV #3 before footshock #4.

This reasoning underscores my previously stated recommendation that "Study 1 needs to show freezing, 22-kHz USV and squeak responses in conjunction with four footshocks for two pre-exposed groups." If the USV calls and ensuing footshock US temporally overlap, it would support delay fear autoconditioning. Conversely, if there is a discernible gap between the USV calls and the following footshock US, it would suggest trace fear autoconditioning.

Lacking this data presentation and in-depth analysis, existing literature on classical conditioning does not seem to support the simultaneous or backward conditioning of shock US and squeak CS, as implied by the current study.

Reviewer #2 (Remarks to the Author):

Dear Julian Packheiser and co-authors,

The revised manuscript "Investigating the mechanistic role of painful self-experience in emotional contagion: an effect of auto-conditioning?" covered all my major concerns and I don't have any questions.

Thanks to the editors for inviting me to carefully assess authors' answers raised by the reviewer #3. I think all 3 major concerns are adequately addressed. Authors added extra experiment in attempts to untangle the CO2 laser from the foot-shock ' effects' on the fear conditioning. They added new discussions to the revised manuscript to cover all concerns. Hence, I think, the revised manuscript can be accepted to the publication.

However, it turns out that this study triggers more additional questions, and follow-up study is necessary. Now I also have additional questions to authors: empathy , i.e. feeling of others, is a great factor which will facilitate fear learning in rodents. In fact, the first demonstration of empathy in mice was shown in the observational fear conditioning phenomenon. If so, authors should keep it in mind since in the current study they played "audible pain-squeaks of others". Others might be a completely random rats, or cage-mates, familiar rats. Authors should design a

new study to address empathy-dependent phenomenon of fear / pain processing.

** See the Nature Portfolio author and referees' website at www.nature.com/authors for information about policies, services and author benefits

Communications Biology is committed to improving transparency in authorship. As part of our efforts in this direction, we are now requesting that all authors identified as 'corresponding author' create and link their Open Researcher and Contributor Identifier (ORCID) with their account on the Manuscript Tracking System prior to acceptance. ORCID helps the scientific community achieve unambiguous attribution of all scholarly contributions. You can create and link your ORCID from the home page of the Manuscript Tracking System by clicking on 'Modify my Springer Nature account' and following the instructions in the link below. Please also inform all co-authors that they can add their ORCIDs to their accounts and that they must do so prior to acceptance.
<https://www.springernature.com/gp/researchers/orcid/orcid-for-nature-research>

Reviewer #1 (Remarks to the Author):

I appreciate the detailed responses and data analyses to address my comments and concerns. However, I remain perplexed by the authors' interpretation of their findings that pre-exposure to footshocks leads to auto-conditioning to self-generated squeaks that can underlie emotional contagion.

Response: We thank the reviewer for the appreciation of our new results and analyses and hope that we can relieve any residual confusion about our data interpretation.

During the shock pre-exposure procedure (which involves four unpredictable 1 sec, 0.8 mA footshocks with an intershock interval of 240-360 sec), each shock US elicits a tightly coupled squeak response. This squeak response spans both audible and ultrasound ranges within the same 1 sec period, as depicted in Fig. 4A, which is essentially a UR. Subsequent to the footshock, 22-kHz USV calls arise within the intershock interval.

Unless I have misunderstood, the authors argue that each squeak acts as a CS undergoing autoconditioning to the shock US. Yet, considering the shock US and its evoked squeak response transpire almost simultaneously (with the sequence being shock US followed closely by squeak CS), how can fear conditioning occur under such simultaneous (backward) conditioning conditions?

Response: The reviewer is correct about the autoconditioning hypothesis we test. The reviewer is also right, that a CS simultaneous with, or shortly following the onset of a US is far from ideal to maximize associations. However, there is literature, which we now highlight in the revised introduction, to show that simultaneous and even backwards conditioning is possible (e.g., Barnet et al., 1991; Prével et al., 2019; Cole & Miller, 1999). Indeed a very recent study provides some insights into the neural mechanisms of such learning in the dopaminergic system (Seitz et al., 2022). We included a new section in the introduction where we discuss previous findings for auto-conditioning of 22 kHz vocalizations. The new section reads:

“Auto-conditioning is thus generally plagued by unfavorable timings of the self-emitted CS relative to the US. Can auto-conditioning thus happen at all? Several studies have found that a CS can gain excitatory value even if it starts after the US, be it in a simultaneous or even backward conditioning configuration (e.g., Barnet et al., 1991; Prével et al., 2019; Cole & Miller, 1999). For a long time, this seemed at odds with the basic tenets of learning theory, that stimuli need to be predictive to trigger learning (for an overview, see Chang et al., 2003). A recent study has started to shed light on the neural mechanisms of learning in situations in which the CS is not temporally predictive of the US, by showing that the dopaminergic system, important for forward conditioning, is also involved in such backward conditioning (Seitz et al., 2022).”

Barnet, R. C., Arnold, H. M., & Miller, R. R. (1991). Simultaneous conditioning demonstrated in second-order conditioning: Evidence for similar associative structure in forward and simultaneous conditioning. *Learning and Motivation*, 22(3), 253-268.

Chang, R. C., Blaisdell, A. P., & Miller, R. R. (2003). Backward conditioning: mediation by the context. *Journal of Experimental Psychology: Animal Behavior Processes*, 29(3), 171.

Cole, R. P., & Miller, R. R. (1999). Conditioned excitation and conditioned inhibition acquired through backward conditioning. *Learning and Motivation*, 30(2), 129-156.

Prével, A., & Krebs, R. M. (2021). Higher-Order Conditioning With Simultaneous and Backward Conditioned Stimulus: Implications for Models of Pavlovian Conditioning. *Frontiers in Behavioral Neuroscience*, 15, 749517.

Seitz, B. M., Hoang, I. B., DiFazio, L. E., Blaisdell, A. P., & Sharpe, M. J. (2022). Dopamine errors drive excitatory and inhibitory components of backward conditioning in an outcome-specific manner. *Current Biology*, 32(14), 3210-3218.

On the other hand, the USV emitted during the intershock interval seems more plausible as a CS autoconditioned to the subsequent shock US. For instance, the USV #1, following footshock #1, might serve as a CS preceding footshock #2 (indicative of forward conditioning rather than backward conditioning). Similarly, USV #2 could precede footshock #3, and USV #3 before footshock #4.

This reasoning underscores my previously stated recommendation that “Study 1 needs to show freezing, 22-kHz USV and squeak responses in conjunction with four footshocks for two pre-exposed groups.” If the USV calls and ensuing footshock US temporally overlap, it would support delay fear autoconditioning. Conversely, if there is a discernible gap between the USV calls and the following footshock US, it would suggest trace fear autoconditioning.

Response: We thank the reviewer for this suggestion. We fully agree with the reviewer that during pre-exposure to shocks 22 kHz USVs indeed provide a suitable cue that the animals could condition to. However, given that we do not include these 22 kHz in our playbacks, such 22 kHz vocalizations are unlikely to explain our results since we do not observe any fear responses during the playback session until the playback of the squeak itself (as evident from the lack of fear responses during the baseline periods in all three experiments). Thus, even if the animals were to condition to 22 kHz vocalizations during shock pre-exposure, they are not exposed to these stimuli during the playback session until they start emitting these USVs themselves as triggered by the presentation of the squeak. We however agree with the reviewer that once the animals start to emit 22 kHz USVs, this behavior might amplify the fear responses initially triggered by the squeak playback as they might also have been auto-conditioned to. We thus added a section to the discussion to highlight this possibility as an explanation for the findings in experiment 1. The new section reads:

“Another limitation of the present study is that we cannot rule out that other cues available for auto-conditioning such as 22 kHz vocalizations might have contributed to the fear responses observed in the first experiment. While the initial fear responses were certainly evoked by the presentation of the playback squeak cue as any baseline period was void of fear responses, the animals started to emit 22 kHz USVs soon after the first playback. As the animals could have auto-conditioned to other fear behaviors during pre-exposure, this could have amplified the fear responses during the playback session.”

Lacking this data presentation and in-depth analysis, existing literature on classical conditioning does not seem to support the simultaneous or backward conditioning of shock US and squeak CS, as implied by the current study.

Response: We hope the new introduction now includes relevant references that show that simultaneous or even backward conditioning is possible, as it has been demonstrated behaviorally and recently also associated with dopaminergic signaling that is strongly linked to Rescorla-Wagner theory. Especially when only exposed to a few trials of conditioning, a backward conditioned CS can gain considerable excitatory value (for an overview, see Chang et al., 2003). Since we only used four shocks during our pre-exposure, this observation fits our experimental paradigm.

Chang, R. C., Blaisdell, A. P., & Miller, R. R. (2003). Backward conditioning: mediation by the context. *Journal of Experimental Psychology: Animal Behavior Processes*, 29(3), 171.

Reviewer #2 (Remarks to the Author):

Dear Julian Packheiser and co-authors,

The revised manuscript "Investigating the mechanistic role of painful self-experience in emotional contagion: an effect of auto-conditioning?" covered all my major concerns and I don't have any questions.

We thank the reviewer for appreciating our revision.

Thanks to the editors for inviting me to carefully assess authors' answers raised by the reviewer #3. I think all 3 major concerns are adequately addressed. Authors added extra experiment in attempts to untangle the CO2 laser from the foot-shock ' effects' on the fear conditioning. They added new discussions to the revised manuscript to cover all concerns. Hence, I think, the revised manuscript can be accepted to the publication.

We thank the reviewer for recommending that the manuscript can be accepted for publication

However, it turns out that this study triggers more additional questions, and follow-up study is necessary. Now I also have additional questions to authors: empathy , i.e. feeling of others, is a great factor which will facilitate fear learning in rodents. In fact, the first demonstration of empathy in mice was shown in the observational fear conditioning phenomenon. If so, authors should keep it in mind since in the current study they played "audible pain-squeaks of others". Others might be a completely random rats, or cage-mates, familiar rats. Authors should design a new study to address empathy-dependent phenomenon of fear / pain processing.

Response: We thank the reviewer for their positive evaluation of our revised manuscript and highly appreciate the additional work of evaluating another reviewer's comments. The reviewer is right in pointing out that the identity of the animal emitting the squeaks was not manipulated in our study in rats. The reviewer is also right that in male mice, the identity of the mice receiving shocks influences the magnitude of the response of the observer, and it would thus be very relevant to test if pain squeaks contain enough information about the identity of the squeaker to modulate the response in male mice. However, through a series of experiments, in which we have systematically manipulated the identity of the rat receiving shocks, we have shown that the same is not true in rats (Han et al., 2019): rats show similar responses to a highly familiar cage-mate receiving shocks, to an unknown rat of the same strain receiving shocks, and even to an unknown member of an unfamiliar strain receiving shocks (Han et al., 2019). Accordingly, the notion that nocifensive behavior in response to witnessing the distress of others reflects an other-regarding response that would be appropriately called empathy is doubtful, at least in rats (see Keyzers et al., 2022 for a detailed analysis of this issue). Based on these prior experiments in rats, it seemed unlikely that varying the identity of the recorded rat would be insightful in rats. However, we still added a paragraph to the discussion reading:

“Finally, it could be interesting to investigate the role of familiarity with the emitter of the squeaks as the present study played back squeaks of unknown rats. Previous research in mice has demonstrated that familiarity during an observation of shock increases the transmission of fear (Gonzalez-Liencrez et al., 2014). However, our own experiments have shown that familiarity does not play a similar role in rats: rats show fear responses even when witnessing unfamiliar rats receive shocks, and higher levels of familiarity do not translate into robustly higher fear responses in observers (Han et al., 2019).”

Gonzalez-Liencrez, C., Juckel, G., Tas, C., Friebe, A., & Brüne, M. (2014). Emotional contagion in mice: the role of familiarity. *Behavioural brain research*, 263, 16-21.

Han, Y., Bruls, R., Soyman, E., Thomas, R. M., Pentaraki, V., Jelinek, N., ... & Keysers, C. (2019). Bidirectional cingulate-dependent danger information transfer across rats. *PLoS biology*, 17(12), e3000524.

REVIEWERS' COMMENTS:

Reviewer #1 (Remarks to the Author):

I appreciate the authors' citations of studies on backward/simultaneous conditioning. Nonetheless, I'd like to underscore several distinctions.

The studies by Barnet et al. (1991) and Prevel et al. (2019) both implemented **second-order** conditioning, characterized by pairings of CS1-CS2 where CS1 had previously been forward paired with a US. This is distinct from the **first-order** backward/simultaneous conditioning (US-CS) utilized in the present study. Further, the Cole & Miller (1999) study showcased backward conditioning but with a 30-s CS duration and, notably, only at a "low" level. It did not manifest at "moderate" or "high" levels. This suggests that such conditioning might be more of an exception than a rule. It is worth noting that this study also indicated effective backward conditioning through a second-order approach.

In addition, the authors-mentioned Seitz et al. (2022) study utilized an **appetitive** paradigm and an average CS duration of 30-s (ranging from 2-58-s).

Given these differences, equating the findings on backward conditioning from these studies with the authors' proposal of a 1-s footshock US paired with a 1-s squeak CS (or, to be exact, a US-induced squeak UR) may not be entirely fitting. A more analogous comparison might be found in fear-potentiated startle and eyeblink (aversive) conditioning studies. These often employ shorter CS durations, similar to the authors' study. Importantly, many of these studies don't necessarily support the concept of backward/simultaneous conditioning.

Therefore, I'd like to reiterate the necessity that Study 1 present data on freezing, 22-kHz USV, and squeak responses, especially in relation to the four footshocks for the two pre-exposed groups.

Reviewer #4 (Remarks to the Author):

The manuscript presents an interesting and thought-provoking exploration into the emotional contagion. The study aims to demonstrate (a) that hearing squeaks from other animals can trigger fear responses in pre-exposed listeners and (b) that a painful experience, when paired with hearing squeaks, enhances these responses. While Experiment 1 successfully supports the first aim, Experiments 2 and 3 yield inconclusive results for the second aim.

Addressing Reviewer #1's Concerns:

The authors have adequately addressed the temporal relationship between the US and CS, substantiated by additional discussion and citations.

Additional Interpretation:

While the authors showed that the stimuli in Experiments 2 and 3 induced sub-squeak pain, the stimuli used may not have been sufficiently intensive to serve as a robust US. Therefore, the lack of conditioning observed could be due to this insufficiency, but it is to be noted that this interpretation does not necessarily negate the auto-conditioning hypothesis. This point could be optionally added to the discussion.

Dear Professor Keyzers,

Thank you for your patience during the peer review process. Your manuscript entitled "Investigating the mechanistic role of painful self-experience in emotional contagion: an effect of auto-conditioning?" has now been seen again by our referees, whose comments appear below. Please note that we recruited a new reviewer (denoted #4) to comment on the rebuttal alongside Reviewer #1 and, in light of their advice, I am delighted to say that we are happy, in principle, to publish a suitably revised version in *Communications Biology* under the open access CC BY license (Creative Commons Attribution v4.0 International License).

Response: We are grateful for the helpful editorial process and are happy that our manuscript has been in principle accepted for publication.

REVIEWERS' COMMENTS:

Reviewer #1 (Remarks to the Author):

I appreciate the authors' citations of studies on backward/simultaneous conditioning. Nonetheless, I'd like to underscore several distinctions.

The studies by Barnett et al. (1991) and Prével et al. (2019) both implemented **second-order** conditioning, characterized by pairings of CS1-CS2 where CS1 had previously been forward paired with a US. This is distinct from the **first-order** backward/simultaneous conditioning (US-CS) utilized in the present study. Further, the Cole & Miller (1999) study showcased backward conditioning but with a 30-s CS duration and, notably, only at a "low" level. It did not manifest at "moderate" or "high" levels. This suggests that such conditioning might be more of an exception than a rule. It is worth noting that this study also indicated effective backward conditioning through a second-order approach.

In addition, the authors-mentioned Seitz et al. (2022) study utilized an **appetitive** paradigm and an average CS duration of 30-s (ranging from 2-58-s).

Given these differences, equating the findings on backward conditioning from these studies with the authors' proposal of a 1-s footshock US paired with a 1-s squeak CS (or, to be exact, a US-induced squeak UR) may not be entirely fitting. A more analogous comparison might be found in fear-potentiated startle and eyeblink (aversive) conditioning studies. These often employ shorter CS durations, similar to the authors' study. Importantly, many of these studies don't necessarily support the concept of backward/simultaneous conditioning.

Therefore, I'd like to reiterate the necessity that Study 1 present data on freezing, 22-kHz USV, and squeak responses, especially in relation to the four footshocks for the two pre-exposed groups.

Response: We thank the reviewer for this in-depth revision of the added literature and have once more amended our manuscript to highlight the potential caveats of backwards conditioning in both the introduction and discussion. They read:

““Several studies have found that a CS can gain excitatory value even if it starts after the US, be it in a simultaneous or even backward conditioning configuration (e.g., Barnett et al., 1991; Prével et al., 2019; Cole & Miller, 1999) although the boundary conditions for simultaneous or backward conditioning to occur are likely more constrained compared to forward conditioning.”

“Given that in auto-conditioning, pain squeaks occur during the US, rather than preceding and predicting the US, ideal conditions for conditioning are not met, and the animals may have undergone a form of pseudo-conditioning⁴³, defined as the emergence of conditioned responses in the presence of a CS that was not paired with the US.”

In our manuscript, we already outlined the temporal relation of freezing, 22 kHz vocalizations and squeaking in relation to the footshocks. This information is presented in Figure 2. We thus feel that we sufficiently complied with the reviewer’s request.

Reviewer #4 (Remarks to the Author):

The manuscript presents an interesting and thought-provoking exploration into the emotional contagion. The study aims to demonstrate (a) that hearing squeaks from other animals can trigger fear responses in pre-exposed listeners and (b) that a painful experience, when paired with hearing squeaks, enhances these responses. While Experiment 1 successfully supports the first aim, Experiments 2 and 3 yield inconclusive results for the second aim.

Response: We thank the reviewer for the evaluation of our manuscript.

Addressing Reviewer #1's Concerns:

The authors have adequately addressed the temporal relationship between the US and CS, substantiated by additional discussion and citations.

Response: We are thankful for the reviewer’s assessment of this point.

Additional Interpretation:

While the authors showed that the stimuli in Experiments 2 and 3 induced sub-squeak pain, the stimuli used may not have been sufficiently intensive to serve as a robust US. Therefore, the lack of conditioning observed could be due to this insufficiency, but it is to be noted that this interpretation does not necessarily negate the auto-conditioning hypothesis. This point could be optionally added to the discussion.

Response: The reviewer is absolutely right that auto-conditioning is not negated by our findings. We now explicated this point and hope that future research in this domain will be conducted. It reads:

“As our results do not negate the auto-conditioning hypothesis, it is important to conduct follow-up studies that go beyond behavioral approaches.”